# Innovative Strategies for Hair Regrowth and Skin Visualization

**DOI:** 10.3390/pharmaceutics15041201

**Published:** 2023-04-10

**Authors:** Qiuying Mai, Yanhua Han, Guopan Cheng, Rui Ma, Zhao Yan, Xiaojia Chen, Guangtao Yu, Tongkai Chen, Shu Zhang

**Affiliations:** 1Guangdong Provincial Key Laboratory of Advanced Drug Delivery Systems, Center for New Drug Research and Development, Guangdong Pharmaceutical University, Guangzhou 510006, China; 2Science and Technology Innovation Center, Guangzhou University of Chinese Medicine, Guangzhou 510405, China; 3State Key Laboratory of Quality Research in Chinese Medicine, Institute of Chinese Medical Sciences, University of Macau, Macau 999078, China; 4Stomatological Hospital, Southern Medical University, Guangzhou 510280, China

**Keywords:** hair regrowth, skin visualization, microneedles, regenerative medicine, natural products

## Abstract

Today, about 50% of men and 15–30% of women are estimated to face hair-related problems, which create a significant psychological burden. Conventional treatments, including drug therapy and transplantation, remain the main strategies for the clinical management of these problems. However, these treatments are hindered by challenges such as drug-induced adverse effects and poor drug penetration due to the skin’s barrier. Therefore, various efforts have been undertaken to enhance drug permeation based on the mechanisms of hair regrowth. Notably, understanding the delivery and diffusion of topically administered drugs is essential in hair loss research. This review focuses on the advancement of transdermal strategies for hair regrowth, particularly those involving external stimulation and regeneration (topical administration) as well as microneedles (transdermal delivery). Furthermore, it also describes the natural products that have become alternative agents to prevent hair loss. In addition, given that skin visualization is necessary for hair regrowth as it provides information on drug localization within the skin’s structure, this review also discusses skin visualization strategies. Finally, it details the relevant patents and clinical trials in these areas. Together, this review highlights the innovative strategies for skin visualization and hair regrowth, aiming to provide novel ideas to researchers studying hair regrowth in the future.

## 1. Introduction

Hair loss is a common disorder in humans and has multiple causes, including aging, autoimmune reactions, and stress [1]. It is reported that about 50% of men and 15–30% of women are affected by hair loss and the psychological burden it carries. Furthermore, its prevalence also seems to be increasing rapidly [2,3]. Based on its causes and symptoms, hair loss can be divided into three types: androgenetic alopecia (AGA), alopecia areata (AA), and other types of hair loss. Of these, the most common is AGA, a chronic and progressive disease that is also called male pattern baldness [4,5]. In AGA patients, dermal papilla cells (DPCs) express high levels of androgen receptors (ARs), which increases their sensitivity to androgens [6]. When the androgen testosterone binds to an AR, it is converted into dihydrotestosterone (DHT) in the cytoplasm of DPCs. This reaction is catalyzed by the enzyme type II 5α-reductase (SRD5A2) [7]. AA is an autoimmune disease clinically characterized by small, bald patches on the head. Several clinical trials have reported the use of Janus kinase (JAK) inhibitors, including ruxolitinib, tofacitinib, and baricitinib, for AA treatment [8]. Commonly, human hair follicles (HFs) undergo cyclic phases, including the anagen phase, catagen phase, and telogen phase. These phases are regulated by hair follicle stem cells (HFSCs). Anagen is a phase of rapid growth in which HFSCs self-renew and produce differentiated progeny cells. It can last for 3–6 months on average. In the catagen phase, which lasts around 2 weeks, hair growth slows down or even stops, and the lower portion of the HF shrinks. However, the club hair remains in place. Thereafter, when the follicle enters the telogen phase, the hair slowly falls off. To re-enter anagen, dermal papillae stimulate the proliferation of their HFSCs. However, if the DPCs are damaged due to any reason, the hair cycle will stop in the telogen phase, and no hair regrowth will occur, eventually resulting in hair loss.

Hair loss causes aesthetic issues, low self-esteem, and social anxiety [9]. Hence, efficient therapies for hair loss are necessary. So far, the drugs approved for hair loss by the US Food and Drug Administration (FDA) are limited to minoxidil (MXD) and finasteride (FIN). MXD needs to be converted into its active derivative, minoxidil sulphate, and this reaction is catalyzed by sulfotransferases [10]. The sulfotransferase 1 (SULT1) family of enzymes is expressed in the lower sheath of the HF and the liver. It converts the prodrug MXD to its active form, minoxidil sulfate, in the outer root sheath (ORS) of HFs. SULT1A1 is the predominant isoenzyme responsible for the sulfonation of MXD in HFs. The expression of sulfotransferase in the scalp is different in different individuals, explaining the heterogeneity in clinical responses to MXD therapy [11]. In some studies, the skin permeability and retention of FIN and MXD were enhanced using liposome-based delivery systems [12,13,14]. For example, the use of FIN has been limited because its systemic administration can cause sexual dysfunction. To solve this problem, DMSO-modified liposomes were prepared for the topical delivery of FIN. The permeation capacity of DMSO could promote FIN delivery to HFs, mitigating the adverse effects of systemic administration [13]. In addition to drug treatment, HF transplantation is another therapeutic option for hair loss. In this treatment, HFs from the back of the head are transplanted to the site of hair loss [15]. However, this treatment is limited by its high cost and the shortage of donors. Hence, it is necessary to explore creative strategies for hair loss treatment. In addition, information on drug accumulation within the skin needs to be obtained via skin visualization. An appropriate imaging technique will demonstrate not only the depth but also the amount of drug accumulation in the skin. In conclusion, innovative treatment strategies combined with appropriate methods of skin visualization could improve therapeutic paradigms for hair regrowth.

Before detailing innovative therapeutic strategies, it is vital to understand the mechanisms of hair regrowth. The mechanisms of hair regrowth depend on two related types of cells, HFSCs and DPCs. Most strategies for hair regrowth target the activation of HSFC or DPC proliferation via specific signaling pathways to promote hair regrowth. We can divide these innovative strategies into three groups: external stimulation, microneedling, and regenerative treatments. External stimulants include light, ultrasound, electric current, and stretch. Except for ultrasound, which can also increase transdermal drug efficiency, others promote hair growth via their own unique mechanisms. Microneedles (MNs) not only act as a vector for drug delivery into the skin but also stimulate re-epithelialization to promote hair regrowth. In addition, regenerative treatment has now become popular for supporting hair regrowth. In this review, we discuss the effect of hair regrowth via regenerative treatments, including biomimetic extracellular matrix, extracellular vesicles, and organoids. By constructing a biomimetic extracellular environment to simulate in vivo hair regrowth, hair regeneration can be enhanced by adjusting the environment of the HF. In addition, exosomes are now the most common agents for regenerative medicine. Exosomes derived from HFSCs or DPCs could have the same effect as their parent cells and thus promote hair regrowth. Moreover, organoids provide HFs for transplantation. As we know, traditional drug treatments for hair loss have various adverse effects. Thus, alternative medicines with minor side effects and high safety are necessary. Natural products have been applied to treat hair loss for a long time in China, showing significant clinical efficacy. Relevant research has been carried out to provide a theoretical basis for the treatment of hair loss. Moreover, skin visualization could assist researchers in understanding the temporal and spatial distributions of drugs in target skin tissues. Hence, some imaging approaches have been introduced in this review to study drug uptake within the skin. With further comprehension of these innovative strategies, patent applications and clinical trials based on external stimulation and regenerative strategies have been conducted.

In this review, we focus on new innovative therapeutic strategies for hair regrowth as well as advanced imaging techniques for skin visualization. In addition, we detail the effect of natural products on hair regrowth.

## 2. Structure of the Skin and Hair Follicles

Understanding the structure of the skin is essential for treating skin diseases. As the largest organ of the body, the skin has distinct functions. It regulates the temperature of the body, impedes the loss of salts and fluids, and prevents pathogen invasion. Typically, the skin is composed of three different layers: stratum corneum (SC), epidermis, and dermis. The SC forms the outermost layer of the skin and is roughly 10–20 μm thick. The SC acts as a barrier, preventing pathogens, such as bacteria and viruses, and toxic substances from entering the body, especially from invaginating dermal regions. Beneath the SC lies the epidermis, which is composed of keratinocytes, melanocytes, Merkel cells, and Langerhans cells. The dermis, located beneath the epidermis, constitutes the largest part of the skin and is made up of lymphatic vessels, collagen, elastin, sebaceous glands, and HFs. The function of the dermis is to structurally support the skin and provide nutritional support to HFs. As a major component of the skin, HFs extend from the skin’s surface to the dermis and are surrounded by sebaceous glands, together creating a special niche [16]. The detailed anatomy of the human and mouse HFs is presented in (Figure 1). HFs consist of a connective tissue sheath (CTS), an inner root sheath (IRS), an outer root sheath (ORS), a hair bulb, and a hair shaft. Hair bulbs are surrounded by dermal fibroblasts, and fibroblasts constitute dermal papillae (DPs), which possess the ability to induce HF renewal. However, how growth factors affect HFs has not been clarified. The IRS is made up of three layers, including the root sheath, Huxley’s layer, and Henle’s layer, and the three-layered structure of the IRS decides the shape of the hair shaft. Outside the IRS lies the ORS, which encompasses the whole IRS and hair shaft. The outermost layer of an HF is the CTS, which consists of several layers of fibroblast cells. Human and mouse HFs have the same basic structure, including the bulge, ORS, IRS, isthmus, and infundibulum. They also possess the same principal cell types and undergo hair cycling, anagen phases, catagen phases, and telogen phases. However, some differences exist between human and mouse HFs. First, the anagen phase of human HFs can last for several years, but the anagen phase of mouse skin HFs only lasts for 2–3 weeks. Second, the markers of epithelial HFSCs are different. CD34, a bulge cell marker in the mouse, is not expressed in human bulge cells but is instead expressed in ORS cells in human HFs [17]. Third, the HF cycle in humans is asynchronous, while that in mice is synchronized [18]. Despite these differences, murine models remain important for studying human HF cycling.

## 3. Molecular Mechanisms of Hair Regrowth

Understanding the mechanisms of hair regrowth is undoubtedly of great significance for developing hair loss treatments. Hair growth strategies target two kinds of cells, HFSCs and DPCs. Hair regrowth can be promoted by increasing the proliferation or migration of these cells. The related molecular mechanisms involving microRNA, the immune system, and signaling pathways are all summarized in (Table 1).

As mentioned earlier, fresh hair growth is driven by HFSCs, which reside in a telogen HF’s lower segment made up of a hair germ (HG) bulge. At the start of the hair cycle, quiescent HFSCs residing in the bulge niche are activated to regenerate progenitor cells and then differentiate into the ORS. These ORS cells exit the bulge niche and lose the expression of the HFSC marker CD34. After completing the growth cycle, the ORS cells return to the bulge niche and remain there until the next hair cycle. The molecular mechanisms of SC fate reversibility involve the suppression of glycolysis and activation of oxidative phosphorylation and glutamine metabolism. In the niche, glutamine metabolism and HFSC reversibility are regulated by the mammalian target of rapamycin complex 2 (mTORC2)-Akt signaling axis. Consequently, the deletion of mTORC2 from the epidermis could lead to a loss of HFSC fate reversibility [25]. Moreover, telogen-to-anagen (T-A) transition involves two processes. The first is HG proliferation, which results in the production of multipotent progenitor cells (MPPs); these cells provide fuel for fresh hair shaft formation. Meanwhile, subsequent bulge SC (BuSC) activation occurs through Sonic Hedgehog (SHH) signaling. 

Micro-RNAs (miRNAs) are single-stranded non-coding RNA molecules and are 18–22 nucleotides in length. They are essential regulators of gene expression in the epithelium. However, the miRNA-mediated regulation of gene expression in the HF mesenchyme remains to be elucidated. A study reported that miR24 is upregulated during skin epithelium differentiation in vitro, acting as a pro-differentiation factor for the epithelium. However, miR24 may play a different role in the epithelium under different physiological conditions in vivo. By analyzing HFSC-enriched miRNAs in HG cells, researchers found that miR24 was significantly downregulated upon hair growth activation, and its expression was inversely correlated with HF progenitor activation. This limited the sensitivity of HF progenitors to growth stimuli. As miRNAs regularly reduce the transcript levels of their targets, gene set enrichment analysis (GSEA) was used to identify three Target Scan-predicted miR24 targets. Plk3, Bbc3, and Trp53imp1 exhibited a significant inverse correlation with miR24 expression. Through a series of Western blot and qRT-PCR tests, Plk3 expression was confirmed to be negatively correlated with miR24 expression both in vitro and in vivo. In short, the results showed that miR24 limits the growth of HF progenitors by targeting Plk3, downregulating the expression of CCNE1, a key cyclin required for cell-cycle entry [22]. Recently, miR122—mainly recognized as a tumor suppressor—was found to be greatly overexpressed in the bulbs of balding HFs in AGA patients when compared with nonbalding ones. Notably, insulin-like growth factor 1 receptor (IGF1R), which is believed to act as an anti-apoptosis factor via the activation of AKT/ERK signaling and is known to be an indispensable morphogenetic and mitogenic regulator, was verified as a target of miR122 and could be repressed by the miRNA. A dual-luciferase assay confirmed that miR122 binds to the wild-type 3′-UTR to suppress the miRNA expression of IGF1R. In conclusion, in AGA, miR122 induces apoptosis in human DPCs (hDPCs) by targeting IGF1R [21]. 

In addition to miR124 and miR122, four other differentially expressed miRNAs have been identified in AGA. These are miR106b, miR125b, miR211, and miR410. However, the detailed function of miRNAs in DPCs has not yet been elucidated [33]. In addition, fibroblasts in DPs, the principal signaling niche, regulate HFSCs. Activating Hedgehog signaling can increase fibroblast heterogeneity in DPs and produce distinct hyper-activated DP states. Single-cell RNA sequencing revealed that mutant fibroblasts contain activated regulatory networks, including Gil1, Alx3, Hoxc8, Ebf1, Sox18, and Zfp239, which upregulate secreted factors involved in hair-growth-related and hair morphogenesis factors. Among these, the TGF-β ligand Scube3 is non-conventional. Fascinatingly, Scube3 is only expressed in growing DPs but not in resting follicles. Microinjection of the SCUBE3 protein is sufficient to induce new hair growth. This indicates that Hedgehog regulates mesenchymal niche function in HFs via a SCUBE3/TGF-β-dependent mechanism [24]. Not only does Hedgehog regulate the mesenchymal niche function of HF, but immune cells also influence the function of stem cells. However, the mechanisms through which immune cells influence HFSC function is currently unknown. Recently, regulatory T cells (Tregs) were identified in the skin around the HF, which holds a major subset of HFSCs. Through lineage-specific cell depletion, the study found that Tregs promote HF regeneration by augmenting HFSC differentiation and proliferation. Tregs expressed high levels of Jagged 1, a Notch ligand family member. The expression of Jag 1 in Tregs could facilitate HFSC function and HF regeneration [28]. In addition, the steroid hormone glucocorticoid has also been found to play a vital role in various biological processes, such as metabolism, inflammation, and development [34,35]. The glucocorticoid receptor (GR) is ubiquitously expressed across different tissues. Glucocorticoid stimulates skin-resident Treg cells to promote HFSC activation and HF regeneration by inducing the expression of transforming growth factor β3 (TGF-β3), which activates Smad2/3 in HFSCs and facilitates HFSC proliferation [26]. This enhancement of HFSC proliferation could promote a resumption of the stem cell state, improving hair growth. 

External stimulation can activate specific signaling pathways to promote hair regrowth via independent mechanisms. A study revealed that photobiomodulation therapy (PBMT) promotes the activation of HFSCs and alleviates HF atrophy. PBMT induces the production of reactive oxygen species (ROS) by activating PI3K/AKT/GSK-3β/β-catenin signaling, promoting HFSC proliferation. Moreover, PBMT increases the HF induction capacity of skin-derived precursors (SKPs) and enhances the secretion and expression of Wnt proteins, which synergistically enhances GSK-3β/β-catenin signaling in HFSCs. This provides a theoretical basis for PBMT-mediated hair loss treatment [30]. Furthermore, mechanical stretch is reported to induce hair regeneration. Through molecular and genetic analyses, researchers found that macrophages are recruited by chemokines and M2 macrophage polarization is induced when proper strain is applied for an appropriate duration. These M2 macrophages release the growth factors HGF and IGF-1, which activate stem cells and promote hair regeneration [32]. Besides PBMT and mechanical stretch, electrical stimulation can also promote hair regeneration. Electrical stimulation increases the HF number to promote hair regeneration by improving the secretion of keratinocyte growth factor and vascular endothelial growth factor, thus alleviating hair keratin disorder [31]. 

Overall, understanding the mechanisms of hair growth is vital for studying hair regrowth, and it can provide a theoretical foundation for innovative strategies for promoting hair regrowth.

## 4. Innovative Strategies for Hair Regrowth

So far, MXD and FIN are the only first-line drugs approved for hair loss treatment by the FDA. However, both MXD and FIN have side effects such as hypertrichosis and sexual dysfunction, which create distress among patients. Thus, topical administration has emerged as the preferred route for hair loss treatment. However, the SC—a barrier protecting the skin—blocks drug permeation to the HF. As a result, the drug cannot reach its optimal treatment concentration at its site of action. With the development of transdermal techniques and an improved understanding of hair regrowth, several convenient and efficient hair regrowth strategies have been crafted. These strategies can be divided into three groups: external stimulation, microneedling, and regenerative treatments. 

### 4.1. External Stimulation

Table 2 and include photostimulation, stretch, ultrasound and electrical stimulation. As shown in Table 2, ultrasound acts as a means of drug delivery, increasing transdermal drug delivery efficiency. Meanwhile, the other types of stimulation activate Wnt pathways to increase the proliferation of HFSCs and accelerate entry into the anagen phage.

#### 4.1.1. Photostimulation

PBMT, also called low-level laser therapy (LLLT), is a creative, convenient, and non-invasive treatment strategy for hair loss. In 1967, Hungarian doctors applied a 694-nm low-intensity ruby laser to study the effect of laser treatment on tumor growth. They found that low-intensity laser irradiation did not produce any tumors. However, to their surprise, it promoted hair growth in the area of irradiation in shaved mice [43]. This study was the first to demonstrate that low-intensity lasers can promote hair regrowth. Subsequently, LLLT became a key strategy for hair loss treatment. In LLLT, a low-level laser or LED light is utilized to stimulate cells and tissues and influence physiological functions, enhance wound healing, promote angiogenesis, and relieve inflammation [38,44,45]. However, the mechanism of these physiological functions has remained somewhat unclear. Recent studies suggest that the biological effects of PBMT are caused by the absorption of light by cytochrome c oxidase, which is a part of the mitochondrial respiratory chain. This accelerates ATP generation by increasing the mitochondrial membrane potential through electron transport [46,47,48]. Meanwhile, after laser treatment, the area of irradiation produces ROS, which activate cell proliferation and wound healing [36]. A study revealed that PBMT induces ROS to activate the PI3K/AKT/GSK-3β signaling pathway in HFSCs, relieving the proteasomal degradation of β-catenin by increasing the expression of Wnts secreted by SKPs. In short, PBMT drives HFSC activation and relieves HF atrophy [30]. Although lasers have been used in alopecia treatment, their at-home usage is limited due to the high energy consumption of lasers and large equipment size [49,50,51,52,53]. To address these drawbacks, researchers have designed high-performance flexible red vertical light-emitting diodes (f-VLEDs) for hair regrowth (Figure 2). The f-VLEDs are made up of 900 micro-LEDs in a passive matrix, which has 50 × 50 μm^2^-sized LED chips. The thickness is only 20 μm because of the use of polymer substrates. Compared with conventional lasers, f-VLEDs consume less energy and do not cause epidermal injury to the skin. In in vivo experiments, the shaved skin of mice was stimulated with f-VLEDs (5 mW/mm^2^) for 15 min every day. After 20 days of treatment, the mice in the experimental group presented with more hair regrowth in the area of irradiation than those in the control group [37]. However, PBMT alone is not sufficient to treat hair loss, as the time required to achieve the effects of conventional drug treatment can be 2 months or more. Thus, in clinical settings, PBMT is usually adopted as an assistive technique in combination with MXD and FIN. 

Interestingly, photostimulation of the eyes can also achieve hair growth via pathways different from skin irradiation. External light stimulation of the eyes has been reported to induce hair regeneration by activating HFSCs. Daily blue light stimulation of the eyes for 10 consecutive days was found to induce fresh hair growth on both sides of the trunk, extending to the center of the back. This was different from the hair regrowth seen from the center of the back, extending to the lateral trunk, following the direct irradiation of the dorsal skin. When irradiation is performed with a longer-wavelength green light, the capacity of hair regrowth induction decreases. After studying the mechanisms of light-accelerated hair regeneration, researchers found that M1-type intrinsically photosensitive retinal ganglion cells (ipRGCs), which are located in the inner layer of the retina, express melanopsin, which conveys signals to the suprachiasmatic nucleus (SCN). Subsequently, Hedgehog signaling is immediately activated via the activation of afferent sympathetic nerves and cutaneous norepinephrine is released. Hedgehog signaling leads to the upregulation of the targets Gli1 and Gil2, which activate HFSCs and promote hair regeneration [54]. Although this mechanism provides a new idea for hair regrow, more studies are required to explore the safety of eye stimulation.

#### 4.1.2. Ultrasound Stimulation

Ultrasound has been widely used for transdermal administration due to its excellent ability to promote the permeation capacity of the skin. The biological effects of ultrasound mainly involved three mechanisms: thermal effects, cavitation, and acoustic streaming. Thermal effects increase tissue temperature due to the absorption of kinetic energy [55]. The cavitation effect is caused by the bubbles produced due to changing pressures in the medium, and it can disrupt endothelial membranes and enhance cell membrane permeability [56]. Acoustic streaming, generated due to unidirectional currents in a fluid, can enhance the cavitation effect through bubbles [57]. Ultrasound is a transdermal approach that can assist with drug delivery to the HF in patients with alopecia. Leveraging the cavitation effect induced by ultrasound, researchers designed MXD (Mx)-coated lysozyme-shelled microbubbles (LyzMBs). The mice treated with Mx-LyzMBs combined with ultrasound were found to have significantly longer hair shafts [58]. In addition to drug delivery, ultrasound can also increase the permeation of proteins. The Cas9/sgRNA riboprotein complex carrier can be successfully delivered to DP cells of HFs via ultrasound (Figure 3). Subsequently, the transfer protein construct (Cas9/sgRNA) can recognize and edit the target gene, recovering hair regrowth [39]. In conclusion, ultrasound is an ideal technique for improving drug permeation in the skin, and it can reduce the required drug dose to a certain extent. However, relevant clinical trials for hair loss treatment based on the ultrasound technique have still not been conducted. With a further understanding of ultrasound, these clinical trials can be conducted in the future.

#### 4.1.3. Electrical Stimulation

Electrical stimulation has been used to treat various skin diseases and for wound healing due to its excellent skin permeation capacity [40,59,60]. It achieves these effects by relieving inflammation and promoting vessel regrowth [59,61,62]. With technological developments, electrical stimulation has also been applied for hair regrowth. The concept of electrotrichogenesis (ETC) has been put forward. ETC can enhance the influx of calcium ions into DP cells via voltage-gated transmembrane ion channels, which can facilitate mitochondrial ATP synthesis, activate protein kinases, and stimulate protein synthesis and cell division [63,64]. To confirm the mechanism underlying the effects of electrical stimulation, researchers fabricated an electrode using a conducting polypyrrole (PPy) and provided electrical stimulation to cultured hDPCs. The hDPCs were then co-cultured with mouse embryonic epithelial cells (mECs) to obtain aggregates, which were then transplanted into the backs of immunodeficient nude mice to examine the potential of HF neogenesis. The result shows that electrical stimulation can improve the trichogenic activity of hDPCs through voltage-gated Ca^2+^ and K^+^ channels and the MAPK signaling pathway [65]. Moreover, alternating current treatment also had a similar mechanism. After treatment with a low voltage and frequency of alternating current, hDPCs successfully showed in vitro proliferation, and Wnt/β-catenin, Ki67, p-ERK, and p-AKT expression was upregulated [66]. In addition to alternating current, micro-current stimulation also increased the expression of various growth factors, including FGF, Wnts, VEGF, and IGF-1 [67]. To improve treatment compliance among alopecia patients, a wearable hair regrowth device was designed (Figure 4). This device consisted of an m-ESD that converts random body motions into stable electric pulses, creating a non-invasive electric field (EF) that stimulates hair regrowth. EF can facilitate calcium influx, enhancing cell proliferation, hair growth factor secretion, and hair regeneration. A series of animal experiments showed that the m-ESD increases follicle density and hair shaft length in SD rats and alleviates hair keratin disorder by improving VEGF and KFG secretion in nude mice [31]. Although a wearable electric device has been used for hair regrowth, the comfort of the wearable devices also needs to be considered. Hence, a hydrogel was utilized to fill the micro gap between wearable devices and human skin, converting it into a tissue-like space that feels soft and moisturized. A functional high-mass permeability and low-impedance hydrogel has been developed for application as a liquid electrolyte on the skin and can form an interface for electrical stimulators and wearable devices [41]. In conclusion, electrical stimulation is a potential technology for hair regrowth. We expect that more convenient and comfortable wearable devices could be developed for treating hair loss in the future.

#### 4.1.4. Stretch Stimulation

In addition to photo and electrical stimulation, stretch stimulation can also induce hair regrowth. Recently, several experiments have proven that stretch stimulation can contribute to a series of biological effects, such as cell proliferation, because it influences cell differentiation and migration and maintains cell homeostasis and tissue repair [68]. However, how mechanical forces affect the skin in vivo remains unclear. To elucidate the mechanisms, researchers established a mouse model in which the effect of stretch on the skin’s epidermis can be studied at a single-cell resolution. Using clonal analysis and single-cell RNA sequencing, the researchers found that stretching induces skin expansion by creating a transient bias in the renewal activity of epidermal stem cells [42]. Interestingly, hair regrowth can be induced after applying a proper strain on the skin of the mouse using a stretch device (Figure 5). The mechanisms of hair regrowth involve the Wnt and BPM-2 pathways. Molecular and genetic analyses revealed that stretch stimulation produces chemokines to recruit macrophages and polarize them to the M2 type. Subsequently, M2 macrophage release HGF and IGF-2 growth factors and activate stem cells to promote hair regrowth [32]. Overall, stretch stimulation has the advantages of small equipment size and high compliance. It has thus attracted increasing public attention. Nonetheless, relevant clinical trials on stretch stimulation have not been conducted yet. This is because the duration of investigation of stretch stimulation for hair regrowth is too short, and the specific molecular mechanisms of how growth factors affect HFs have not been clarified.

In conclusion, external stimulation can be seen as an innovative approach to hair regrowth, and several relevant animal experiments have been conducted. However, the experimental data from human studies are limited due to frequency and safety concerns. In addition, wearable devices represent a promising strategy for hair regrowth due to their convenience. More wearable devices should be combined with external stimulation to treat hair loss. 

### 4.2. Microneedles for Hair Regrowth

MNs are advanced transdermal drug delivery tools that have attracted the attention of a large group of researchers. Compared with conventional transdermal strategies, MNs can deliver drugs in a painless and minimally invasive manner. In addition, MNs are low-cost and allow self-administration. Based on the characteristics of MNs, they have been applied for the treatment of various conditions, including melanoma, alopecia, psoriasis, and other skin diseases. MN devices consist of numerous miniaturized needles. The length of an MN is only 100 to 1000 μm. Thus, MNs can pierce the SC easily. MNs are divided into five categories according to their characteristics: solid MNs, hollow MNs, coated MNs, dissolving MNs (DMNs), and hydrogel-forming MNs. Among these, DMNs are the most common for alopecia treatment because of their excellent biocompatibility and convenient administration. Recently, numerous MN-based therapies have been developed for alopecia. Valproic acid-encapsulating DMNs were applied for the first time in alopecia. As an anticonvulsant drug approved by the FDA, valproic acid (VPA) can stimulate hair regrowth effectively by upregulating the Wnt/β-catenin pathway. Carboxymethyl cellulose (CMC), a biocompatible and biodegradable polymer, forms the main matrix material of DMNs, provides the mechanical strength required to pierce through the SC and improves the VPA delivery capability. DMN-VPA has a significantly higher penetration capability and diffuses at a surprisingly faster rate than the topically applied drug. Moreover, it also stimulates wound re-epithelialization signals involved in HF regrowth. Additionally, DMN-VPA induces hair regrowth by upregulating relevant proteins involved in the Wnt/β-catenin pathway, including alkaline phosphatase, proliferating cell nuclear antigen, and loricrin. Compared with the topical application of VPA, the administration of VPA via DMNs induced higher levels of HFSC markers, such as keratin 15 and CD34 [69]. To improve the mechanical strength of MNs, MN patches based on keratin have been applied for hair loss treatment. Recently, a study showed that UK5099, a small molecular drug, can accelerate HFSC activation via glycolytic metabolism. To improve the transdermal efficiency of this drug, MNs made up of UK5099 and hair-derived keratin, including mesenchymal stem cell (MSC)-derived exosomes, were developed. Keratin, the main component of hair, acted as the matrix material. Keratin has a high content of intramolecular disulfide bonds and can thus mimic the network structure of hydrogels via mild and simple disulfide shuffling, enhancing the mechanical strength of MNs as well as allowing sustained drug release capacity and long-term biocompatibility. Through a series of in vitro and in vivo tests, direct MN-mediated exosome and UK5099 delivery to HFs and sustained drug release were confirmed. Of note, combined administration could induce pigmentation and hair regrowth more effectively than the control treatment. This combination could thus provide a useful strategy for hair loss therapy [9]. With the advancement of nanotechnology, nano-enzymes have been brought into the spotlight. In recent years, nanomaterials with enzyme-mimicking activities have been extensively employed in antibacterial and antioxidant therapy. Ceria nanozyme (CeNZ)-integrated MNs are reported to reshape the perifollicular microenvironment in AGA (Figure 6). CeNZs, with their enzyme-mimicking activities, can alleviate oxidative stress in the perifollicular microenvironment for AGA treatment. Researchers designed CeNZ-integrated MNs (Ce-MNs) that can scavenge ROS and promote angiogenesis around HFs to facilitate hair regrowth. Ce-MNs are made up of dissolvable hyaluronic acid (HA) and CeNZs, which can enhance the mechanical strength of Ce-MNs and deliver them to the epidermis and dermis effectively. In vivo tests demonstrated that the Ce-MNs group achieved hair regeneration faster and with a comparable quality of regenerated hair at a lower administration frequency, without any skin damage. On a mechanistic level, Ce-MN treatment alleviated the expression of DHE and 4-HNE and promoted the expression of angiogenesis-related growth factors such as CD31 [70]. Therefore, MNs can act as a skin delivery vector for loaded drugs, proteins, and cell-derived exosomes. Accordingly, MNs can not only improve the efficiency, accuracy, and effectiveness of drug delivery but also stimulate re-epithelialization to promote hair regrowth. However, the injury caused by microneedles cannot be ignored. Transient pain and mild erythema are commonly reported as adverse effects of microneedles [71]. Further, differences in HF length between mice and humans also need to be considered. Human HFs are significantly larger, reaching lengths of 5 mm, whereas mouse follicles are only 1 mm long [17]. Microneedles can provide effective drug delivery to mouse HFs due to their length. However, microneedles may not deliver drugs to the target after application on human skin because their length may be too short to reach the HF site.

### 4.3. Regenerative Medicine for Hair Regrowth

In patients with hair loss, regenerative medicine utilizes stem cells or tissue engineering to provide clinical hair regrowth strategies. In this part, we will introduce regenerative treatments using three agents: biomimetic extracellular matrix, extracellular vesicles, and organoids. A biomimetic extracellular matrix could expand HSFCs in vitro, and extracellular vesicles derived from HSFCs or DPCs have the same induction capacity as their parent cells. Finally, cultured organoids not only serve as a model for studying hair loss but also provide a promising approach to producing HFs for hair transplantation. Hence, regenerative treatments have immense potential for treating hair loss.

#### 4.3.1. Biomimetic Extracellular Matrix

Regenerative medicine is a promising strategy for the treatment of hair loss. Autologous HFSCs and DPCs can be transplanted into the area of hair loss to promote hair growth. However, the large size of HSFCs precludes them from expanding in vitro because of the absence of the HF extracellular matrix. Hence, the reconstruction of the natural extracellular matrix is a crucial part of hair regeneration. With the development of tissue engineering and regenerative medicine, researchers can now utilize biomaterials to mimic HF microenvironments and amplify multipotent HFSCs and DPCs. This has become a promising treatment strategy. Layer-by-layer (LbL) self-assembly is a single-cell nanoscale surface modification technique applied in drug delivery, targeted gene therapy, and biosensors. Unlike conventional techniques involving a mixture of cells and hydrogel, LbL assemblies can provide control over mechanical properties. Multilayered films made up of polysaccharides and proteins are a promising tool for constructing nanoscale biomimetic cellular microenvironments. Alginate derived from algae is a natural polysaccharide, and gelatin is widely used for ECM tissue engineering because of its excellent biocompatibility and biodegradability. Gelatin is also used as an LbL material because its constitution is similar to that of native ECM collagen. In a recent study, negatively charged alginate and positively charged gelatin were coated onto HFSCs using the LbL self-assembly technology to construct nanoscale biomimetic ECM for HFSCs (Figure 7). TGF-β2 was used as the model drug and loaded into the coating layer to examine its mechanism of action on HFSCs. Through a series of in vitro experiments and in vivo reconstitution assays, TGF-β2 was found to induce the transformation of CD34^+^ stem cells into highly proliferating leucine-rich repeat-containing G-protein-coupled receptor 5+ (Lgr5+) stem cells, which differentiate into all cell lineages found in the hair shaft and inner root sheath to induce hair regrowth. In short, TGF-β2 coated by an LbL film effectively induced an activated state in stem cells, thus mimicking the microenvironment of regular stem cells for tissue regeneration during HF cycling [72]. In addition to HFSCs, LbL self-assemblies that encapsulate DP cells using nanogels have also been developed. DPs, a cluster of highly specialized mesenchymal cells, are located in the lower part of the HF. They have their own unique gene and protein expression profile as well as the capacity to induce hair regrowth. However, traditional 2D cultures gradually lose their DPC inductive ability during passaging. To improve the DPC inductive capacity, 3D cultures are more appropriate as they provide cell-ECM interactions, which could expend DPCs and maintain their inductive properties. Recently, researchers utilized type A gelatin, alginate, and Ca^2+^ to create DP spheroids that mimic the natural intercellular structures of HFs. In this system, the carboxyl group of alginate provides the required negative charge for the cellular encapsulation of LbL self-assemblies. Further, the alginate can be ionically cross-linked following exposure to Ca^2+^ to induce cell aggregation into cellular microspheres. DP spheroid constructs show a native morphology that is consistent with that of primary DPs. Furthermore, analysis for markers such as ALP, Versican, and NCAM showed that high-passage (P8) DPCs retain their relevant hair induction potential, and new HFs are regenerated successfully when the DP spheroids are implanted in vivo [73]. However, cells in the center of the 3D spheroids typically undergo progressive necrosis, which influences the characteristics and functions of these cells and causes the low efficiency of transplantation in vivo. Vascularization has emerged as a potential strategy to address this critical issue. Vascularization can establish sufficient blood perfusion after implantation and also promote communication between co-cultured cells. Human umbilical vein endothelial cells (HUVECs) can influence the function of DPCs. In one study, researchers utilized the LbL technique to construct nanoscale ECM for DPCs. The LbL-DPCs were co-cultured with LbL-HUVECs to construct vascularized DP spheroids. The interactions between DPCs and HUVECs highly mimicked the in vivo DP niche to restore the transcriptional signature of natural DPCs and accelerate HF formation. This strategy realized reproducible, highly regenerative HFs for clinical application and established a foundation for studying the regulatory mechanisms between endothelial cells and DPCs [74]. LbL technology enabled mechanical support for HFSCs and DPCs, creating a similar state in the in vivo cellular microenvironment. These studies demonstrate that constructing a biomimetic environment is a crucial part of hair regeneration and could serve as a novel avenue for hair-regenerative therapy. 

#### 4.3.2. Extracellular Vesicles

DPs are widely considered central to the induction of hair cycling through a paracrine signaling mechanism. DPCs secrete Wnts and TGF-β, causing the proliferation of the matrix and enabling their differentiation into hair fiber and follicular root sheath cells. Extracellular vesicles (EVs) play a crucial role in regulating cell-to-cell communication. Recent studies reported that EVs derived from stem cells exhibit similar functions as their parent cells. Hence, DP-derived EVs (DP-EVs) have the same effects as DPCs, and they can thus promote hair regrowth by activating Wnt/β-catenin signaling. However, the instability and low retention of EVs in vivo have limited the development of EV-based treatments. To improve the retention and stability of EVs, DP-EVs were encapsulated into oxidized sodium alginate (OSA) to form injectable microgels. Compared with alginate hydrogels, OSA hydrogels degraded more rapidly. As a result, DP-EVs were released more quickly and were absorbed by hair matrix cells and HFs. The OS-EVs were confirmed to promote the proliferation of hair matrix cells and prolong anagen to enhance hair growth in vivo [75]. However, the clear mechanism of action underlying the effect of DP-derived EVs has not been understood. Nevertheless, some data suggest that the upregulation of the β-catenin signaling pathway is involved in this process. A recent study found that miR218-5p plays a vital role in exosome-mediated hair regrowth (Figure 8). Compared with 2D DP-derived exosomes, 3D spheroid-derived exosomes express higher levels of miR218-5p. DP-derived exosomes overexpressing miR218-5p accelerate the onset of anagen and downregulate the ENT signaling inhibitor SFRP2, creating a positive feedback loop to promote β-catenin signaling for HF development [23]. In addition to DP-derived EVs, the autologous adipose-derived stromal vascular fraction (SVF) has also been found to be effective in alopecia treatment. SKPs, originating from the adult mammalian dermis, are capable of differentiating into various lineages, including dermal and mesodermal cells. Hence, they show great potential as stem cells for hair regeneration. The SVF is made up of endothelial cells, pericytes, and immune cells, which have neovascular factors that are effective for combatting alopecia. In one clinical study, nine patients with AGA were divided into two groups. The first group received a single transplantation of autologous SVF on a hairless scalp, while the other served as the control group. After 3 and 6 months of treatment, the hair density in the SVF-treated group was significantly higher than that in the control group, and the keratin score was also improved. These results show that SVF treatment is a potential management strategy for alopecia [76]. However, like DPCs cultured in vitro, SKPs also gradually lose their proliferation and induction capacity during 2D culturing. However, 3D co-cultured systems can now restore the proliferation and induction capacity of SKPs. In a study, SKPs were co-cultured with adipose-derived stem cells (ASCs) and epidermal stem cells (Epi-Scs) to imitate the environment of SKPs in vivo. In the 3D co-cultured system, amphiregulin (AREG) made SKPs re-enter anagen earlier, and highly efficient HF reconstitution was achieved through the PI3K and MAPK pathways in vitro and in vivo. These results also indicated the value of AREG as a treatment agent for alopecia [77].

Finally, exosome therapy has brought new hope for persons with hair loss. It does not require any drugs, avoiding any drug-related adverse effects. Moreover, it is a non-surgical treatment and does not cause damage or affect daily life. However, the induction capacity and in vitro retention of exosomes will need to be addressed for their adoption in clinical settings.

#### 4.3.3. Organoid Culture

At present, hair transplantation is still the main treatment for hair loss. However, most patients do not have enough follicles for successful transplantation. Pluripotent stem cells (PSCs) and organoids could provide a new approach to attaining more HFs. Cultured organoid systems consisting of epidermal and dermal cells have been used as skin models to study disease development in vitro for over 40 years. Reciprocal epithelial–mesenchymal interactions are a crucial part of HF formation and the hair cycle. Hence, skin organoids are also used to study hair regeneration. Mice are a standard model for studying the development of skin disease. However, there are various differences between mouse and human HFs. For example, the anagen phase is only 2–3 weeks in mice but can last for several decades in humans. Further, markers of epithelial HFSCs are also different. Furthermore, the mechanism underlying cell fate specification during human skin development is still not understood. Thus, it is necessary to culture organoids with human PSCs. A group of researchers provided step-by-step guidance for the generation of skin organoids via the direct differentiation of PSCs [78]. A chemical environment conducive to the induction of the skin’s epidermal and dermal cell layers was created by adding modulated signaling factors, including TGF-β inhibitor and BMP4, to induce the formation of a surface ectoderm. Consequently, the skin organoid system could act as a suitable model for studying the fate of the HF cycle as well as the pharmacological effects and genetic factors involved in the hair cycle at the anagen stage. In addition, skin appendages are vital for skin organoids. However, it is difficult to construct skin organoids with growing functional skin appendages. To solve this problem, researchers used PSCs to induce the differentiation of epidermal and dermal cells and generate skin organoids with skin appendages. A 3D integumentary organ system (3D-IOS) dependent on the embryoid body (CDB) method was constructed using iPS cells based on an in vivo transplantation model (Figure 9). The CDB explant had a boundary region with multiple embryonic stem cells, including various types of epithelial and mesenchymal tissues, which provided developing organ-forming fields. Before transplanting multiple EBs in vivo, Wnt10b treatment was administered to stimulate EB clustering. After 30 days, the transplanted EBs had successfully differentiated into a 3D-IOS system, which included mature HFs and sebaceous glands. The 3D-IOS system could be removed and transplanted into wounds, exhibiting complete physiological function in vivo [79]. Apart from utilizing the CDB method to create a space for embryonic stem cell aggregation, TGF-β and FGF could also be used to co-induce neural crest cells and epithelial cells to create a skin organoid system. After incubation for 4–5 months, a skin organoid composed of a stratified epidermis, fat-rich dermis, and HF with sebaceous glands could be generated. Meanwhile, sensory neurons and Schwann cells constructed a network that targets Merkel cells in the organoid HF, mimicking the touch circuitry in humans. Through a series of detection methods, including single-cell RNA sequencing and comparison with fetal specimens, the skin organoids were found to be equivalent to human facial skin. Following grafting in nude mice, the skin organoids could form planar hair-bearing skin [80]. Though organoids have attracted attention from researchers, PSC-derived skin organoids have various limitations, including the lack of critical cell populations such as sweat glands, blood vessels, immune cells, and off-target cell lineages. This needs to be studied in greater depth.

## 5. Skin Visualization

Topical administration on the skin not only prevents potential systemic side effects but also provides sustained drug release. However, the depth of drug permeation into different skin layers is unknown. To assess drug penetration and retention in specific skin structures, skin visualization techniques are needed. These techniques can provide images of drug accumulation in the skin. Indeed, some skin visualization techniques can also quantify the amount of drug accumulation. These imaging techniques are specific and sensitive and provide qualitative or quantitative information using invasive or non-invasive approaches. Each imaging technique has its own features and limitations. Some key skin visualization techniques are summarized in Table 3. The appropriate imaging methods are chosen depending on the properties of the agent to be detected.

In various skin diseases, it is vital to understand the spatial and temporal distribution of drugs in target tissues and organs. An appropriate imaging method could assess the uptake of drugs ex vivo and in vivo. Radioisotopes have been widely used for studying absorption into the skin. In autoradiography, radioisotopes not only provide information on localization in specific skin areas but also possess high sensitivity at a subcellular level. With the help of these radiographic techniques, information on the uptake of drugs can be obtained. However, there are some limitations, as the technique requires radiolabeling and is time-consuming. Hence, alternative imaging methods are required. With the development of optical instruments, optical imaging—as a non-invasive imaging way—has attracted interest. The most commonly used optical instrument is confocal microscopy, a non-invasive imaging technique that allows us to visualize the permeation of drugs into skin layers at a high resolution. However, the high equipment cost and the complicated image interpretation limit its wide-scale use.

Most optical microscopy methods are limited by photon scatter. Hence, the visible spectral region in the images cannot be beyond a hundred micrometers. Compared with optical microscopy, optical coherence tomography (OCT), used for imaging tissue morphology, can improve the penetration depth to the 1–2 mm range. Although the depth of imaging improves, it is insufficient to study the fine structure of the skin and enable morphological assessment of the skin and blood vessels. Recently, ultra-wideband raster-scan optoacoustic mesoscopy (RSOM) has shown potential in examining different layers of skin and the structure of the microvasculature at a high resolution. A key limitation hindering the detection of skin features and the quantification of drugs is the accurate identification of the skin’s boundary. RSOM uses a 3D surface segmentation method to identify the skin’s boundary (Figure 10). Even over discontinuities and diffuse interfaces, this approach allows us to localize the skin’s boundary through dynamic programming. This generates faster and more reproducible results than manual segmentation [88]. RSOM not only provides a high resolution and contrast as an optical imaging system, but it also has the capacity of tissue penetration across a few millimeters. Though RSOM has not yet been applied in alopecia, its function of depicting the microvasculature in superficial tissue is particularly suitable for studying the microvasculature of the HF microenvironment as well as the melanin layer. Thus, it could be used to assess hair growth and the hair cycle. 

Apart from understanding the microvasculature of skin, understanding the cutaneous penetration pathways of drugs is also essential. Mass spectrometry imaging (MSI) can provide details regarding drug spatial distribution and penetration. Nowadays, the common techniques used to visualize skin penetration are confocal Raman microscopy, infrared microscopy, and confocal laser scanning microscopy (CLSM), which capture an image of the skin and visualize the penetration of drug molecules. However, instead of quantitative information, these imaging techniques only provide qualitative outcomes to characterize the spatial distribution of the drug within the skin. Recently, researchers used desorption electrospray ionization (DESI)-MSI to study the spatial cutaneous distribution of a topical agent, as well as the corresponding skin structures, to image endogenous skin constituents and gain insights into drug penetration routes (Figure 11). As DESI-MSI does not require sample preparation, the skin tissue could be analyzed directly, and preliminary insights about molecular spatial distribution could be gained with the help of ultra-high-performance liquid-chromatography (UHPLC)-MS/MS analysis. For example, econazole nitrate (ECZ) creams and a micelle formulation based on D-α-tocopheryl succinate polyethylene glycol 1000 (TPGS) were examined in bioequivalent porcine skin. The DESI-MSI images showed the spatial distribution of ECZ and TPGS in 40-μm-thick horizontal sections, and TPGS was mainly found in the upper epidermis (<80 μm). Indeed, through the co-localization of drugs and endogenous skin elements, DESI-MSI could provide relevant insights into drug penetration pathways [85]. 

In addition to drug penetration pathways, the characterization of the molecular structure of the barrier layer, the SC, is necessary for improving diagnostics, allowing drug delivery, and preventing environmental damage. Recently, a state-of-the-art 3D OrbiSIMS technique was used to conduct in situ analysis of ex vivo human skin tissue and to reveal the detailed molecular chemistry of the skin (Figure 12). The instrument used to perform this technique consisted of a traditional time-of-flight secondary ion mass spectrometer (ToF-SIMS) and Orbitrap mass spectrometer, which could provide significantly better biological analysis. The examination of the depth profiles revealed that different classes of compounds exhibited different patterns of depth penetration in the skin. For example, the intensity of phospholipid ions was minimal within the SC until the transition into the epidermis, marking the SC–epidermal boundary. Ceramides are only localized within the SC. After crossing the SC–epidermal boundary, the ceramides reached a baseline level. Through in situ depth analysis of skin samples, the components of surface sebum and the intrinsic lipid matrix could be identified, which were adapted for human and porcine skin. This technique could not only be used for endogenous species and molecules but could also be used to demonstrate the permeation of peptides into the human SC [87]. 

In conclusion, it is vital to track and quantify the uptake of drugs and relevant compounds into the skin. In alopecia, the most common imaging technique is confocal microscopy, which is always used to detect the permeation of drugs into the skin. For example, rhodamine is loaded into MNs to detect their insertion capacity. Through the confocal images of MNs, the permeation of drugs within different skin layers can be visualized. However, most advanced imaging techniques, including photoacoustic imaging, have not yet been applied for alopecia. With a further understanding of skin visualization, numerous scientists are focused on improving the speed and sensitivity of optical imaging tools for non-invasive use in patients.

## 6. Claims for Hair Regrowth

Nowadays, different countries have different regulations and laws for hair regrowth products. In China, hair regrowth products are classified as cosmetics. These products need to be reported to the State Food and Drug Administration (SFDA), and an approval number needs to be obtained. In Japan, hair regrowth products are classified as cosmetic, quasi-pharmaceutical, and pharmaceutical products according to their composition, claims, and efficacy. However, hair regrowth products are classified as pharmaceutical products in the USA and are subject to FDA approval, which encompasses the product name, active ingredient, and concentration. For example, rogaine—a star hair product classified as a medicine in the USA—contains MXD as its main active ingredient. In China, MXD is classified as a drug that can be bought in pharmacies but cannot be used in cosmetics. To gain FDA approval, hair regrowth products must provide substantial data supporting their efficacy for hair regrowth. When using hair regrowth products, both the instructions for application/use and side effects must be noted carefully.

## 7. Natural Products for Hair Regrowth

The prevalence of hair loss seems to be increasing at present, and its age of onset appears to be decreasing. There is no doubt that hair loss has become a worldwide problem. Currently available conventional therapies based on synthetic drugs are still imperfect and have a number of limitations [89]. MXD is approved by the FDA as a hair loss drug and applied topically, but its applications are limited and short-lived due to its unpredictable efficacy and adverse effects. This has led to an increased interest in alternative treatments for hair loss that have fewer side effects, such as formulations containing herbs and/or their active constituents. Therefore, alternative medicines with minor side effects and high safety are the need of the hour. Traditional Chinese medicine has broadened the ideas for treating hair loss. Communities believe that herbal agents come from ‘natural’ plants; thus, they are ‘naturally safe’ without any adverse effects. Natural products represent the wisdom of the ancient Chinese millennia, and they are still used today. Natural products have several benefits, such as fewer side effects, wide-ranging sources, low price, and a greater number of applications. Western medicine focuses on symptomatic treatment, but natural products aim at the fundamental treatment of diseases. Moreover, preclinical studies and clinical trials have also demonstrated the effectiveness of some plant-derived active ingredients for hair growth.

Natural products can regulate micromolecular activity by influencing the expression of inflammatory mediators, inhibiting apoptosis, regulating gene expression and immune function, and thereby reducing HF damage and ultimately preventing hair loss. Some natural plant-derived products have proven effective in preventing hair loss. Some examples of plants used for treating hair loss are detailed in (Table 4).

Quercitrin (quercetin-3-O-rhamnoside) is a natural flavonoid widely found in the flowers, leaves, and fruits of various plants and was reported to be effective for hair regrowth. One study investigated the molecular mechanisms of quercitrin action. Quercitrin prompted HFs to enter the anagen phase and promoted cell proliferation, upregulating the expression of Bcl-2 and Ki67. In addition, quercitrin also increased the mRNA and protein expression of growth factors such as bFGF, KGF, PDGF-AA, and VEGF. The molecular mechanism underlying the effects of quercetin on hair regrowth has been explored preliminarily. Its effect on DP cells has proven that it can promote hair regeneration. Unfortunately, in vivo studies have not been conducted [103].

Astragaloside IV is a cycloartane triterpene saponin. Its effects on hair loss have been investigated. One study reported the mechanistic effects of Astragaloside IV on apoptotic signaling in HFs within the dorsal skin of depilated C57BL/6 mice. These results demonstrated that Astragaloside IV blocks the Fas/Fas L-mediated apoptotic pathway, downregulating Bax and p53, upregulating Bcl-2 and Bcl-xL, activating NF-κB, and inhibiting IkB-α phosphorylation. In summary, Astragaloside IV inhibits apoptosis-regression catagen in HFs via the Fas/Fas L-mediated cell death pathway. Thus, it may be used for the treatment of hair diseases such as alopecia, effluvium, and hirsutism [111]^.^

Epigallocatechin-3-gallate (EGCG), a major compound from green tea, is an anti-cancer agent and antioxidant. In addition, EGCG is used for the treatment of hair loss as a reductase inhibitor [109]. Cellular studies have shown that EGCG increases cell proliferation and decreases cell death by altering the miRNA profiles in HDP cells and sequentially affects paclitaxel-mediated hair loss [112]. In vitro experiments indicated that EGCG promotes hair growth by upregulating phosphorylated Erk and Akt and by increasing the Bcl-2/Bax ratio [108]. Animal experiments show that topical EGCG administration can also prevent hair loss by reducing the T-induced apoptosis of follicular epithelial cells and provoking hair regrowth after epilation [109]. In short, EGCG could have potential in the treatment of AGA.

Preclinical studies are a key component of research on the effect of herbs and their active constituents on hair growth. In comparison to the high number of in vitro and in vivo preclinical studies, there are few clinical trials describing the influence of herbs and their active constituents, as well as that of polyherbal formulations on hair growth. A small clinical study conducted with three volunteers demonstrated that 10% EGCG can promote hair growth by prolonging the anagen phase [108]. In one case series of 14 patients with AA, Morita et al. reported that the topical administration of the immunotherapy agent squaric acid dibutylester (SADBE) for a mean of 6.9 months led to no or poor hair regrowth. However, the subsequent administration of a combination of topical SADBE treatment plus oral CEP for a mean duration of 7.6 months resulted in satisfactory hair regrowth in six of the 14 cases [92]. Meanwhile, attempts have been made to develop new pharmaceutical dosage forms. Given that cedarferol is known to treat hair loss, a cedarferol nanoemulsion and cedarferol cream were applied to the skin in an animal model and were found to increase drug permeability and significantly promote hair regrowth [98,99].

As a material base for the medicinal effects of natural products, the active ingredients of natural products can be derived from a wide range of sources and have a clear classification of efficacy. Natural products and active ingredients have high safety and remarkable efficacy. However, at present, research on the prevention of hair loss using active ingredients from natural products has certain limitations. For example, some studies have only assessed individual factors or proteins. Other related functional proteins and complete signaling pathways are relatively simple to study, and the specific mechanisms need to be examined. Notably, most of the basic research on natural product extracts or active ingredients focuses on cell or animal hair loss models, and systematic clinical trials are lacking.

## 8. Patent Applications and Clinical Trials for Hair Loss Treatment

In recent years, basic topical administration treatments aimed at reducing hair loss have developed rapidly. Most studies have focused on safety, efficiency, and prospective clinical application. To provide a further understanding of the development trends in hair loss treatment, the recent clinical trials and patent applications have been summarized in Table 5 and Table 6. The patents that are relevant for photostimulation and exosome treatment have been included. 

As shown above, patent applications for some advanced strategies have been submitted. For example, as part of patent number CN202210069596.5 [113], human hair outer root sheath cells (HHORSC)-derived exosomes can be used to induce DP cells for hair regrowth. In the study, the exosomes derived from HHORSCs (HHORSCs-Exo) upregulated the expression of ALP, versican, and α-SMA by 2.1-, 1.7-, and 1.3-fold, respectively, compared with the control treatment. This demonstrates their induction capacity for DPCs. In addition, relevant photostimulation devices have also been the subject of patents. Patent number CN107899139 describes the application of LED-red light (610–650 nm) for the treatment of alopecia. In the related study, mice were randomly divided into two groups: a control group and a red-light treatment group. After 1 month of treatment, the treatment group had almost no alopecia, but the control groups showed obvious alopecia. The results revealed that red light of the wavelength 610–650 nm could prevent hair loss, providing a unique strategy for relieving the adverse effects of chemotherapeutic drugs causing hair loss. Many new low-level laser technologies that claim to support hair regrowth have been released commercially. Laser and light therapy act as a novel, convenient, drug-free, and non-invasive treatment strategy to activate hair anagen and promote hair growth. Numerous studies have shown that laser and light therapy can effectively treat different types of hair loss in various animal models without causing side effects. Recently, trials for light therapy have also been initiated. For example, the REVIAN study evaluated the efficacy and safety of the REVIAN System in male participants with AGA. The REVIAN System includes a Cap configured for portable use with a rechargeable battery and adapter and active LEDs for modulated light therapy (MLT trademark). Daily 10 min treatments are performed at home over the course of 26 weeks.

**Table 5 pharmaceutics-15-01201-t005:** Recent patents for hair regrowth treatments.

Classification	Patent/Application Number	Patent Title	Features	Assignee	Filling Year	Status
Equipment	CN201711166676.8 [114]	Application of LED-red light with 610 nm–650 nm wavelengths in alopecia treatment	A 610 nm and 650 nm LED red light can effectively prevent hair loss	Harbin Medical University	2018	Granted
	CN202010582452.0 [115]	Equipment for laser therapy on androgenetic alopecia and alopecia treatment methods	The laser sterilizes the area of hair loss	Liruiya	2021	Granted
Regenerative medicine	CN202210069596.5 [113]	A method of HHORSCs exosomes promoting dermal papilla cells to induce hair regeneration	Promotes DPCs to induce hair regrowth	Guangdong LIYI Technology Co. Ltd.	2022	Granted
	US17693457 [116]	Methods and compositions for aesthetic and cosmetic treatment and stimulating hair growth	Increases skin volume to prevent or treat hair loss and related diseases	Pluristem Ltd.	2019	Granted
	PCT/IB2021/062226 [117]	Use of miRNA-485 inhibitors for inducing hair regrowth	Increases hair density, follicle density, and hair shaft thickness	Biorchestra Co. Ltd.	2022	Granted
Drug	US17562976 [118]	Hydrazone amide derivative and application thereof in the preparation of medicaments for preventing and treating alopecia	\	Shenzhen Cell Inspire Pharmaceutical Dev Co. Ltd.	2022	Granted

**Table 6 pharmaceutics-15-01201-t006:** Recent clinical trials for hair loss treatment.

Study Title	Medication	ClinicalTrials.gov Identifier	Status	Study Date
ENERGI-F701 for Female Hair Loss Treatment	Drug: ENERGI-F701	NCT03351322	Phase 2	May 2018–December 2019
Safety and Efficacy of HST 001 in Male Pattern Hair Loss	Biological: HST 001–0.1 mL X 20 injections	NCT04435847	Phase 1	May 2020–January 2021
Modulated Light Therapy in Participants With Pattern Hair Loss	Device: REVIAN 101 Device: REVIAN 102 Device: REVIAN 103 Device: REVIAN 100	NCT04019795	Phase 3	January 2017–May 2019
Adipose-derived Stem Cell Conditioned Media as a Novel Approach for Hair Regrowth in Male Androgenetic Alopecia	Combination Product: Non-concentrated adipose-derived stem cell conditioned media and 5% Minoxidil Combination Product: Concentrated adipose-derived stem cell conditioned media and 5% Minoxidil Combination Product: Placebo and 5% Minoxidil	NCT05296863	Phase 3	October 2021– December 2021
Androgenetic Alopecia Treatment Using Varin and Cannabidiol Rich Topical Hemp Oil: A Case Series (Hair Regeneration)	Drug: Hemp Oil	NCT04842383	Early Phase 1	April, 2021–October 2021
Topical Cetirizine in Androgenetic Alopecia in Females	Topical cetirizine	NCT04481412	Phase 2 Phase 3	Ongoing since July 2020
A Study Evaluating the Efficacy and Safety of SM04554 Topical Solution in Male Subjects With Androgenetic Alopecia	Topical SM04554 solution	NCT03742518	Phase 2 Phase 3	November 2018–December 2021
Safety, Tolerability and Pharmacokinetics of KX826 in Healthy Male Subjects With Androgenetic Alopecia Following Topical Single Ascending Dose Administration	KX0826	NCT04984707	Phase 1	January 2019–October 2019
Safety and Efficacy Study of Topical DLQ01 in the Treatment of Androgenetic Alopecia (AGA) in Men	Drug: prostaglandin F2a analogue in vehicle solution high dose Drug: prostaglandin F2a analogue in vehicle solution low dose Drug: active ingredient-free vehicle solution to DLQ01	NCT05636904	Phase 1 Phase 2	December 2022–March 2024
Topical Cetirizine 1% vs Minoxidil 5% Gel in Treatment of Androgenetic Alopecia	Drug: Cetirizine Drug: Minoxidil	NCT04293822	Phase 4	June 2020–November 2021

## 9. Conclusions and Outlook

More and more people are suffering from hair loss, and the focus on hair quality has increased. Conventional treatments for hair loss include drug therapy and hair transplantation. However, both approaches have low efficacy for hair regrowth. Thus, advanced strategies are necessary. First, it is vital to gain a better understanding of the mechanisms underlying hair regrowth. To date, numerous mechanisms of hair regrowth and relevant pathways and specific molecular signals have been uncovered. Wnt/β-catenin pathways and Hedgehog signaling promote HFSC proliferation and enhance hair regrowth. Some special molecules, such as miRNAs, upregulate or downregulate relevant growth factors to stimulate hair regrowth. Treg cells also play a role in hair regrowth. Together, these factors influence HFSCs and DP cells to achieve hair growth. Thus, studying how HFSCs or DPCs are regulated by various growth factors under physiological or diseased states can reveal new therapeutic targets for hair loss. Nowadays, the topical application of drugs creates various advantages for hair loss treatment. However, the SC creates a barrier to transdermal drug delivery. Some external stimulants, such as light, ultrasound, electrical current, and stretch, can assist with drug delivery across the SC to HFs without causing severe damage to the skin. However, most of these approaches require large and expensive equipment. In the future, we hope that external stimulation can be combined with artificial intelligence to design new wearable devices that are convenient and portable for hair loss therapy.

Regenerative medicine appears to be a promising alternative for hair loss treatment. In addition to HFSCs, DP cells also have the capacity for hair regeneration due to the interaction between epithelial and mesenchymal niches. DPC-derived exosomes and LbL assemblies of HFSC microspheres have been applied to induce hair regrowth. However, the low retention of exosomes in HFs remains a problem and limits the development of regenerative treatments. Hydrogels are an excellent vehicle for prolonging the retention time of exosomes derived from DPCs in HFs. Moreover, HF organoid culture systems can achieve HF formation and hair growth in vitro, providing new ideas for studying hair regrowth mechanisms.

Though creative strategies could promote hair regrowth, the first-line drugs for hair loss treatment are still MXD and FIN. Both of these have severe adverse effects. Thus, natural products have attracted attention in this area. Compared with synthetic agents, natural products have various advantages, such as minor side effects and lower costs of treatment. Furthermore, the most important feature of natural products is the multiple biochemical actions of diverse plant extract combinations that can help in treating hair loss. However, we should be aware that only some active ingredients of natural products penetrate the epidermal barrier and reach the HF, and the desired therapeutic effect could influence hair growth by stimulating or inhibiting specific growth factors and modulating signal pathways. Thus, it is necessary to combine transdermal techniques to achieve the targeted delivery of active ingredients at the required site and achieve healing efficacy. MNs seem to be suitable candidates for the delivery of natural products for hair loss treatment. The active ingredients of natural products could be loaded into MNs to construct relevant delivery systems. The MN-based delivery system would not only cross the SC barrier but also stimulate the epidermis to promote hair growth. However, some issues must be focused on. In natural products, the synergistic action of the active ingredients is key for treating hair loss. However, the plant components responsible for the main hair loss-reducing effects remain unknown. This also contributes to the side effects of natural products. Therefore, in the future, it will be necessary to identify the exact components of natural products that support hair regrowth and elucidate their mechanisms of action. Meanwhile, different provinces have different requirements for natural products. Thus, the relevant laws and regulations should be standardized throughout the country.

Finally, understanding the delivery and diffusion of topically administered drugs in human skin is crucial in hair loss research. Different skin imaging methods can characterize and quantify the spatial distribution of drugs within skin ex vivo and in vivo. Although various imaging methods have been applied to the skin, the imaging methods used for studying hair loss are so far limited. The most common imaging method is confocal microscopy, which is usually used to detect the depth of drug permeation. However, photoacoustic imaging could be an ideal method to visualize angiogenesis in cases of hair loss, allowing us to estimate the stage of the hair cycle based on the levels of angiogenesis. Overall, hair loss is worth studying. If we take additional time to learn from HFs and understand the processes of repair and regeneration, more and more potential strategies for hair regrowth will be discovered. Fresh and exciting strategies for hair loss management will be imminent in the future.

## Figures and Tables

**Figure 1 pharmaceutics-15-01201-f001:**
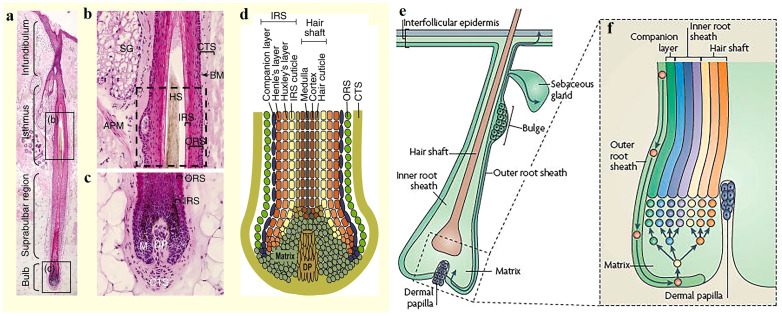
(**a**–**d**) Human hair follicles (HFs). (**a**) Sagittal section of an HF from the human scalp (anagen VI) showing permanent (infundibulum and isthmus) and anagen-associated (suprabulbar and bulbar area) components. (**b**) High-magnification image of the isthmus. (**c**) High-magnification image of the bulb. (**d**) Schematic illustrating the concentric layers of the outer root sheath (ORS), inner root sheath (IRS), and shaft in the bulb [19]. Copyright 2008 Elsevier Science. (**e**,**f**) Mouse HFs. During normal conditions, bulge stem cells (SCs) are periodically activated to form a new HF (see part (**e**)). As bulge SC progeny move down the ORS to the base of the HF, they subsequently become matrix cells. After proliferation, these cells differentiate along one of seven hair lineages (see part (**f**)) [20]. Copyright 2022 Springer Nature.

**Figure 2 pharmaceutics-15-01201-f002:**
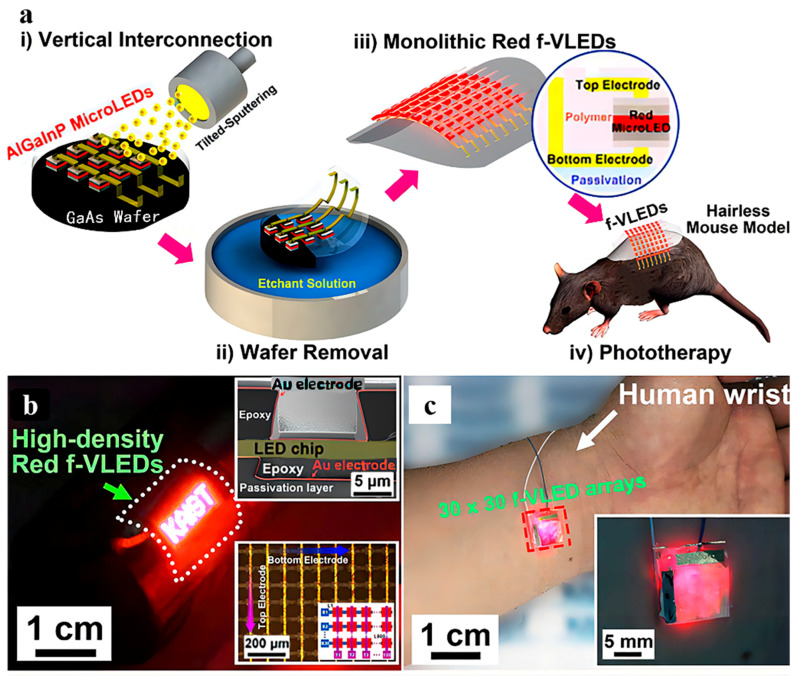
(**a**) Schematic illustration of the fabrication procedure and trichogenic photostimulation by monolithic flexible AlGaInP vertical LEDs. (**b**) Optical image of monolithic f-VLED with the red word “KAIST” under a bending state. The upper inset represents a cross-sectional SEM image of monolithic red f-VLEDs. The lower inset shows a magnified microscopic image of 30 × 30 f-VLEDs with a 50 × 50-µm^2^ chip. (**c**) Photograph of red f-VLED array affixed on the surface of a human wrist. The inset shows a magnified image of the 30 × 30 LED array [37]. Copyright 2018 American Chemical Society.

**Figure 3 pharmaceutics-15-01201-f003:**
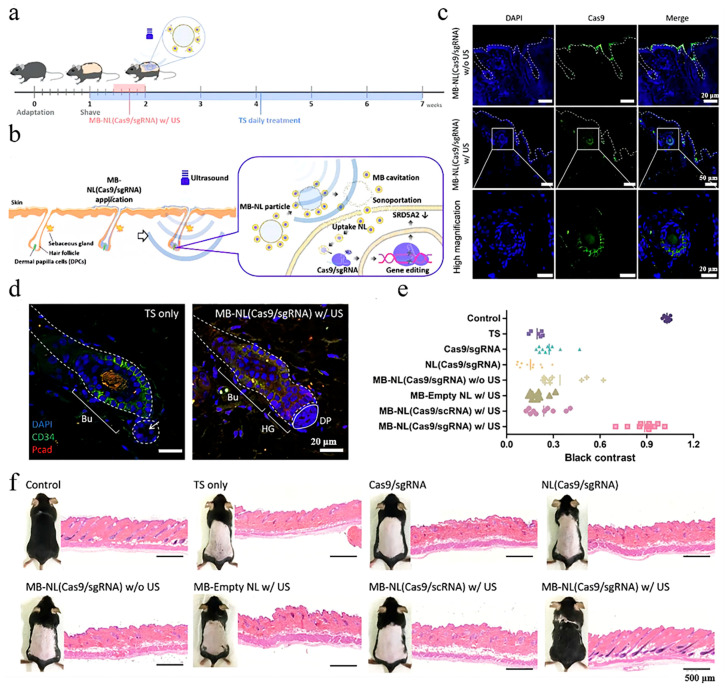
Ultrasound-activated particles as a CRISPR/Cas9 delivery system for androgenic alopecia therapy. (**a**) Schematic diagram of the in vivo experiment. (**b**) Schematic diagram of the US flash exposure (MI = 0.61) and enhancement of NL. (**c**) Confocal laser scanning microscopy (CLSM) images of the mouse skin harvested at 2 h after treatment with MB-NL(Cas9/sgRNA) particles. (**d**) CLSM images of the mouse skin and HFs stained with anti-CD34 (green) and anti-P-cadherin (red) antibodies after TS- or US-activated ML-NL(Cas9/sgRNA) treatment. (**e**) Comparison of black hair contrast intensity. (**f**) H&E staining of harvested mouse skin after 7 weeks of treatment [39]. Copyright 2020 Elsevier.

**Figure 4 pharmaceutics-15-01201-f004:**
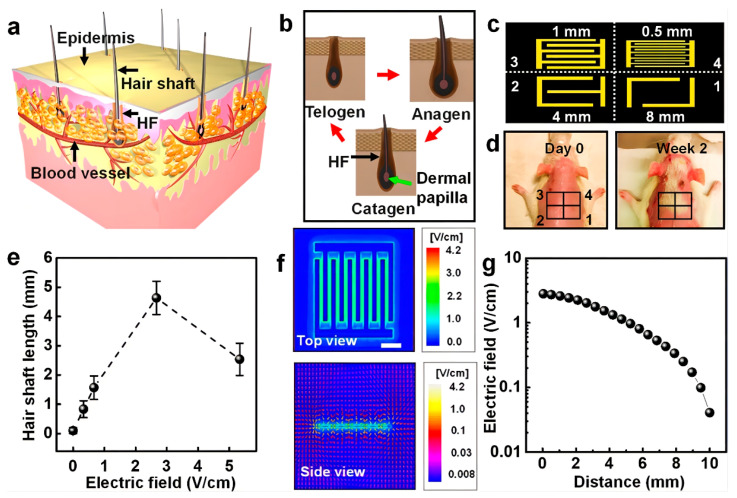
Hair regeneration effect in SD rats under stimulation with the m-ESD. (**a**) Schematic illustration of HFs in the skin. (**b**) Histomorphology of the hair cycle, including anagen, catagen, and telogen phases. (**c**) Schematic diagram of a series of interdigitated electrodes (1–4) with different gap widths. (**d**) Photographs of rats with removed hair (day 0, **left**) and after 2-week treatment (**right**). (**e**) EF-stimulated hair shaft length as a function of the EF intensity (n = 6). (**f**) Top and side views of EF distribution (gap width = 1 mm) after stimulation with a voltage of ±150 mV. (**g**) EF strength at different distances perpendicular to the plane of the interdigitated electrode (gap = 1 mm). All data in (**e**) are presented as mean ± s.d [31]. Copyright 2019 American Chemical Society.

**Figure 5 pharmaceutics-15-01201-f005:**
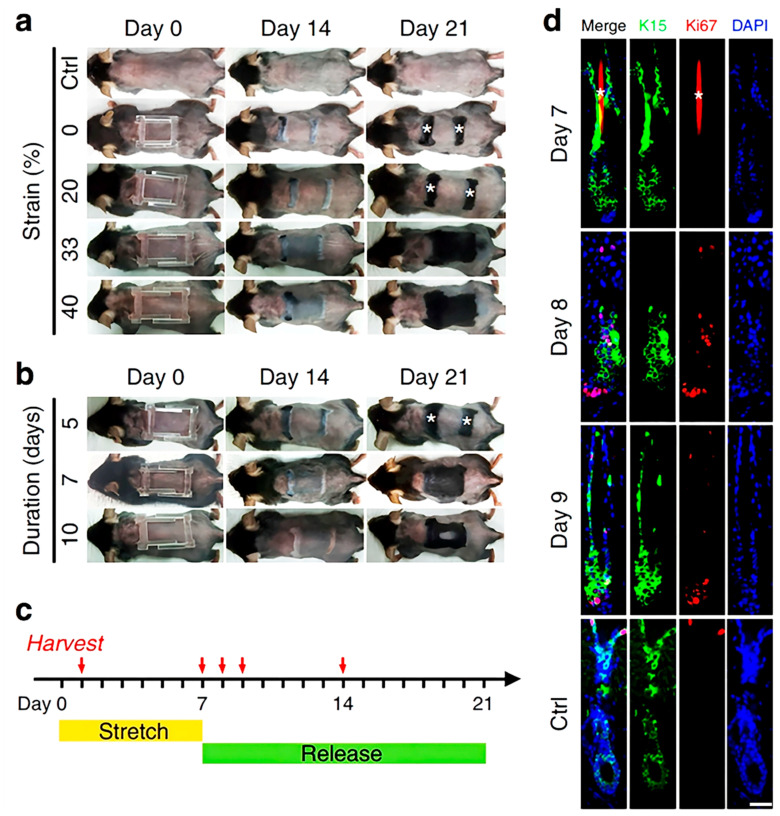
Stretch-induced hair regeneration is dependent on strain and duration. (**a**) Stretch for 7 days under different amounts of strain. * Anagen initiated in glue-fixed area due to hair plucking when removing the skin-stretching device. (**b**) Stretch under 33% strain for different durations. (**c**) Schematic showing the optimal stretching conditions and sample collection times. Red arrows mean the time of harvest mice. (**d**) Dual immunostaining for K15 and Ki67 revealed that K15^+^ hair stem cells began to proliferate on days 8 and 9 when stretch was released. * Autofluorescence of hair shafts. Scale bar = 50 μm [32]. Copyright 2019 Nature.

**Figure 6 pharmaceutics-15-01201-f006:**
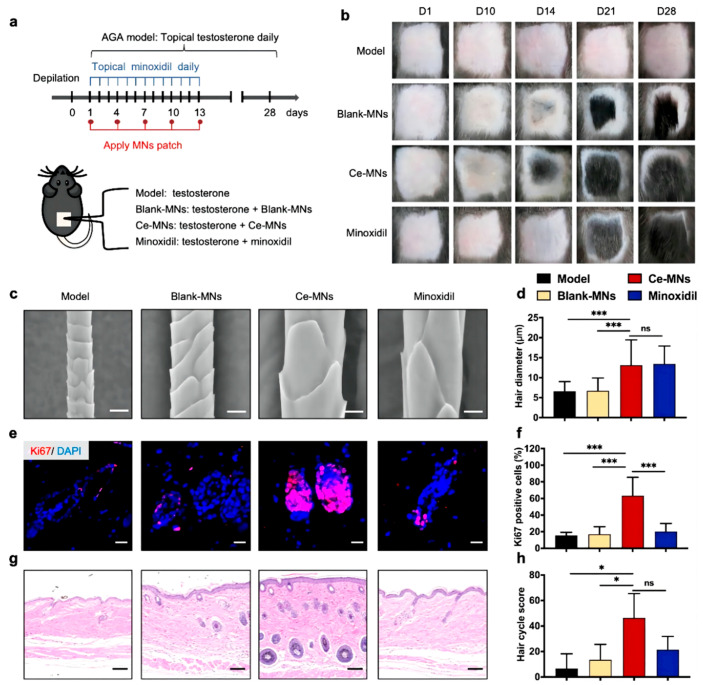
In vivo AGA treatment efficiency of the Ce-MN system. (**a**) Schematic representation showing the establishment of the AGA mouse model via the topical application of a testosterone solution daily for 28 days and the therapeutic strategies in the established model. (**b**) Representative photographs of mouse hair regrowth status in the model, Blank MN, Ce-MNs, and minoxidil groups. (**c**) SEM images of regenerated hair at day 28 post-depilation. Scale bar = 10 μm. (**d**) Diameter of regenerated hair at day 28 post-depilation. (**e**) Representative images of Ki67 expression in skin tissues from different groups at day 10 post-depilation. Scale bar = 20 μm. (**f**) Quantitative analysis of Ki67-positive cells at day 10 post-depilation. (**g**) Hematoxylin-eosin (H&E) staining of the treated skin at day 10 post-depilation. Scale bar = 100 μm. (**h**) Hair cycle score at day 10 post-depilation. All results are presented as the mean ± SD. ns, nonsignificant (*p* > 0.05), * *p* < 0.05, *** *p* < 0.001. [70]. Copyright 2021 American Chemical Society.

**Figure 7 pharmaceutics-15-01201-f007:**
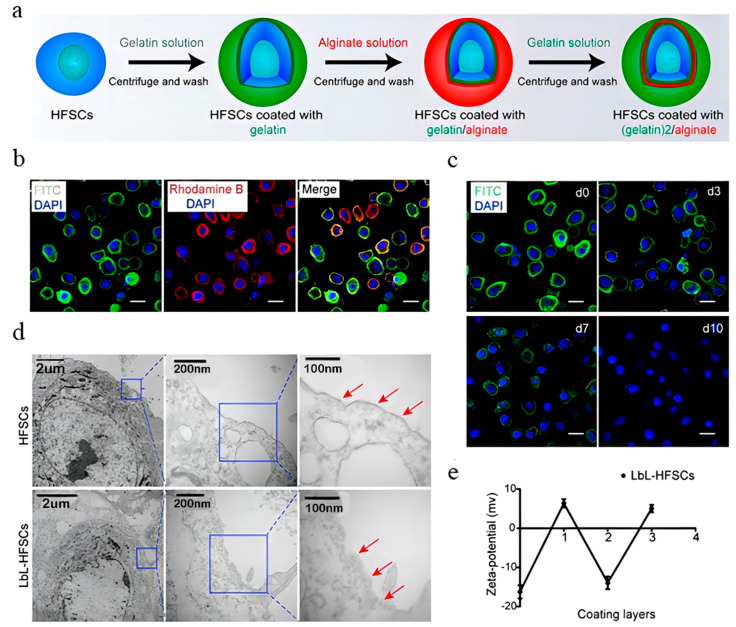
Generation of layer-by-layer−HFSCs (LbL-HFSCs) using the LbL cell coating technique. (**a**) Schematic illustration of LbL-HFSC fabrication using gelatin (green) and alginate (red) to coat HFSCs. (**b**) Confocal laser scanning microscopy (CLSM) images demonstrating an LbL coating on HFSCs; the cell surface was coated with gelatin−FITC (green) and alginate-rhodamine B (red). The cells were in suspension, and nuclei were stained with DAPI (blue). Scale bars: 10 µm. (**c**) Persistence of the biomaterial coated with LbL on the cell surface as shown by CLSM images. Scale bars: 10 µm. (**d**) Transmission electron microscopy (TEM) images showing the comparison of uncoated and coated HFSCs. The arrows indicate coating materials on the cells’ surface. (**e**) Corresponding changes in zeta potential depending on the different layers of coating on the cell surface [72]. Copyright 2020 Ivyspring International.

**Figure 8 pharmaceutics-15-01201-f008:**
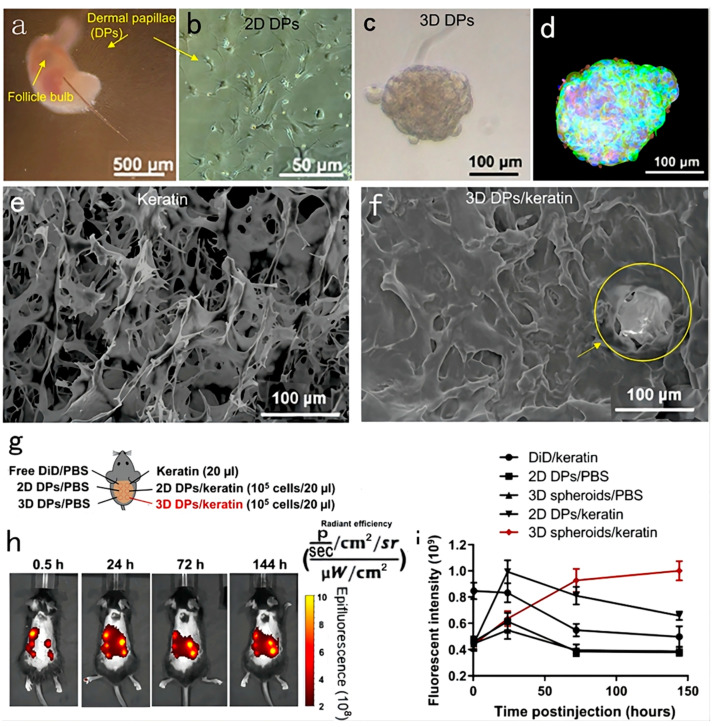
Preparation and characterization of 3D DP spheroids. (**a**) Isolation of mouse DP cells from vibrissae. (**b**) Conventional culture enables the growth of 2D DP cells. (**c**) Growth of DP spheroids in ultralow cell culture flasks. Scale bar, 100 μm. (**d**) Double staining for CD133 (green) and β-catenin (red) in spheroids. (**e**,**f**) Scanning electron microscopy (SEM) images of keratin (**e**) and 3D spheroid-loaded keratin. (**f**) One obvious spheroid is highlighted in yellow. (**g**) Schematic illustrating the injection sites on the back of a mouse for the cell retention study. (**h**) The mouse was shaved and received injections of different formulations on the dorsal skin, as illustrated in (**g**). Cells were labeled with DiD and then resuspended in PBS or keratin for intradermal injection. In vivo imaging system (IVIS) images were taken at different time points. (**i**) Quantification of IVIS images [23]. Copyright 2020 Wiley.

**Figure 9 pharmaceutics-15-01201-f009:**
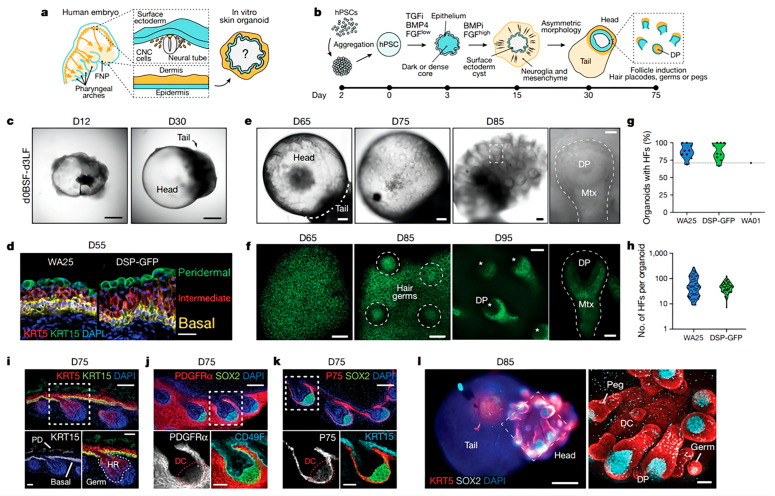
Co-induction of surface ectoderm and CNC cells leads to the generation of hair-bearing skin. (**a**,**b**) Overview of study objectives (**a**) and the skin organoid protocol (**b**). (**c**) Bright-field images of WA25 aggregates on day 12 and day 30 in optimized culture. Scale bars = 500 μm. (**d**) Immunostaining for KRT5+KRT15^+^ basal and KRT15^+^ peridermal layers at day 55. Scale bar = 50 μm. (**e**) Representative bright-field images of HF induction in WA25 skin organoids at days 65–85. Scale bars = 100 μm (left three panels); 25 μm (far right). (**f**) Representative maximum-intensity confocal images (endogenous DSP-GFP) of HF induction in DSP-GFP skin organoids at days 65–95. Scale bars = 100 μm (third panel); 50 μm (second panel); 25 μm (far left and far right). (**g**) Violin plots showing the frequency of HF formation in WA25 and WA01 cultures. (**h**) Violin plots showing the average number of HFs formed per organoid in WA25 and DSP-GFP cultures at days 75–147. (**i**–**k**) Immunostained day-75 WA25 skin organoid with hair placodes. Scale bars (**i**–**k**) = 100 μm (top); 50 μm (bottom). (**l**) Whole-mount sample of day-85 WA25 skin organoid with head and tail structures Scale bars = 250 μm (**left**); 50 μm (**right**) [80]. Copyright 2020 Nature.

**Figure 10 pharmaceutics-15-01201-f010:**
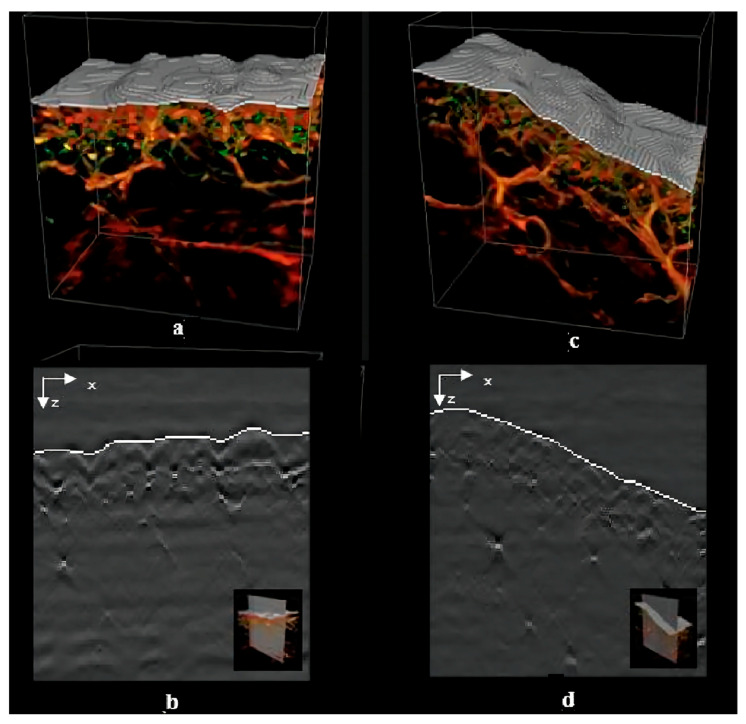
Examples of successful skin segmentation from RSOM data. (**a**,**c**) 3D volumes including the estimated skin layer. (**b**,**d**) 2D slices of the volumes in panels (**a**,**c**), respectively, including the estimated skin contour [88]. Copyright 2020 IEEE.

**Figure 11 pharmaceutics-15-01201-f011:**
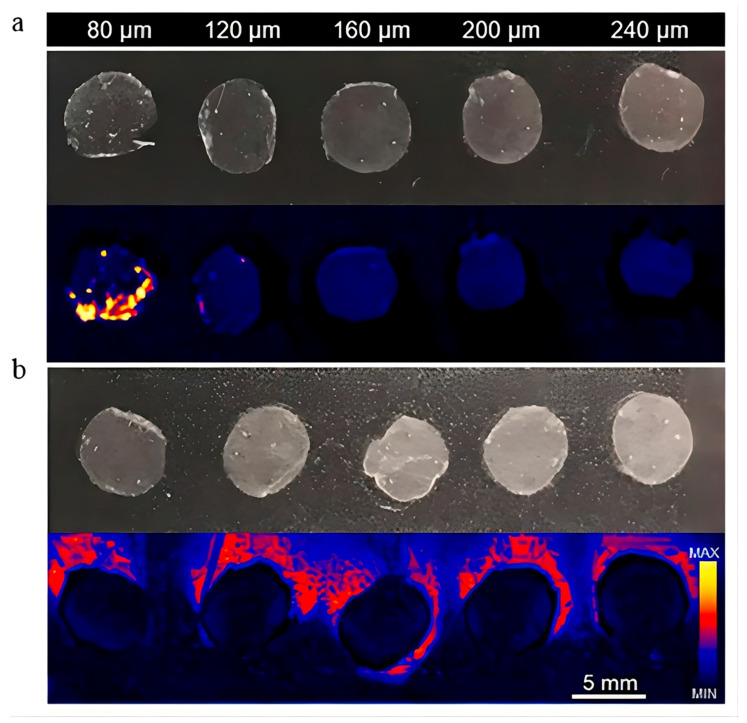
Optical images and DESI mass spectrometry images of horizontal skin sections (40 µm) showing TPGS (*m*/*z* 772.4706) deposition from (**a**) a 4% TPGS micelle solution and (**b**) untreated skin sample [85]. Copyright 2021 Elsevier.

**Figure 12 pharmaceutics-15-01201-f012:**
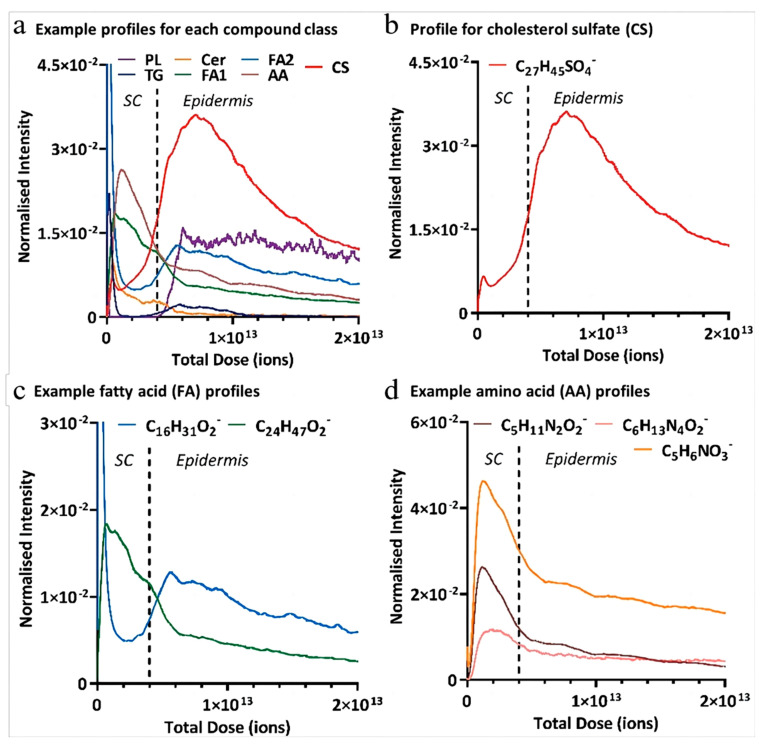
The 3D OrbiSIMS negative polarity depth profile data showing the ion intensity variation as a function of ion dose/skin depth for various putatively assigned compounds. (**a**) Examples of ions from each identified compound class are as follows: PL (phospholipids; C_41_H_77_NPO_8_), TG (C_55_H_101_O_6_), Cer (C_44_H_86_NO_5_), FA1 and FA2 (FAs 1 and 2; C_27_H_47_O_2_ and C_16_H_31_O_2_), AA (amino acids; C_5_H_11_N_2_O_2_), and CS (cholesterol sulfate; C_27_H_45_SO_4_). (**b**) Cholesterol sulfate molecular ions (C_27_H_45_SO_4_). (**c**) FAs, palmitic acid (C_16_H_31_O_2_), and lignoceric acid (C_24_H_47_O_2_). (**d**) NMF compounds, the amino acid arginine (C_6_H_13_N_4_O_2_), and amino acid derivatives PCA (C_5_H_6_NO_3_) and ornithine (C_5_H_11_N_2_O_2_) [87]. Copyright 2022 National Academy of Sciences.

**Table 1 pharmaceutics-15-01201-t001:** Summary of the molecular mechanisms of hair regrowth.

Factors	Molecular of Action	Action Site	Results	Mechanisms	References
microRNA	miR122	Hair follicle	Induces hDPC apoptosis	Repression of IGF1R	[21]
	miR24	Hair follicle progenitors	miR24 limits the sensitivity of hair follicle progenitors to growth stimuli	Downregulation of PIK3 and CCNE1	[22]
	miR218-5p	HFs	Accelerates the onset of anagen	Downregulates SFRP2 to promote β-catenin expression	[23]
Hedgehog signaling	Lepr	DPCs	Induces new hair growth and hair multiplication	Downstream SCUBE3-TGF-β signaling	[24]
mTORC2-Akt signaling	\	Hair follicle stem cell niche	Allows progenitors to return to the hypoxic niche Resumes the stem cell state	\	[25]
Immune system	Glucocorticoids	Treg cells	Facilitates HFSC activation and hair follicle regeneration	GR and Foxp3 cooperatively induce TGF-β3 to activate Smad2/3 in HFSCs	[26]
	Corticosterone	DPCs	Prolongs HFSC quiescence Keeps hair follicles in an extended resting phase	Inhibits Gas6	[27]
	Regulatory T cells	HFs	Augments HFSC proliferation and differentiation	Induces high levels of the Notch ligand family member Jagged 1	[28]
Others	Sirt7	HFSCs	Facilitates the onset of the hair cycle	Promotes NFATc1 degradation	[29]
Photobiomodulation therapy	β-catenin	HFSCs	Drives quiescent HFSC activation Alleviates hair follicle atrophy	Induces ROS and activates the PI3K/AKT/GSK-3β signaling pathways to inhibit β-catenin degradation	[30]
Electrical stimulation	\	HFs	Increases hair follicle number	Improves the secretion of VEGF and KGF	[31]
Mechanical stretch	\	HFSCs	Activates stem cells and promotes hair regeneration	M2 macrophages release growth factor HGF and IGF-1	[32]

**Table 2 pharmaceutics-15-01201-t002:** Summary of external stimulation for hair regrowth.

External Stimulation	Mechanisms	Effects	Characteristics/Advantages	Limitations	References
Light (Photostimulation)	Increasing the mitochondrial membrane potential Producing ROS to activate cell proliferation and wound healing	Activates the anagen phase and the proliferation of hair follicles	No thermal/inflammatory tissue damage	High energy consumption Large equipment size	[36,37,38]
Ultrasound	Thermal, cavitation, and acoustic streaming	Delivers drugs into dermal papilla cells	Low cost, non-invasive, good penetration	Thermal damage	[39]
Electric current	Activating Wnt/β-catenin and MAPK pathways and the secretion of various growth factors	Improves hair follicle density and increases hair shaft length	Low-cost, stable, and small equipment size	Skin damage	[40,41]
Stretch	Activating Wnt and BPM-2 pathways Recruiting Macrophages and inducing the secretion of various growth factors	Activates hair follicle stem cells	Small equipment size and high compliance	\	[32,42]

**Table 3 pharmaceutics-15-01201-t003:** Summary of skin visualization.

Imaging Technique	Depth	Principle	Features	Drawbacks	Applications	References
Optical coherence tomography	<2 mm	Continuous-wave infrared laser radiation	Qualitative Non-invasive	Low resolution, insufficient for observing cell morphology	Vasculature information enables the assessment of inflammatory skin diseases	[81]
Confocal laser scanning microscopy (CLSM)	200 μm	\	Qualitative	Limited to molecules with a fluorophore	Observation of cellular structures	[82]
Ultrasonography	10 cm	\	Qualitative High-resolution	\	Location of vessels	[83]
Mass spectrometry imaging (DESI)-MSI	50–200 µm	Ionization technique	High-throughput Minimum sample preparation Quick results	Lower spatial resolution and sensitivity	Quantitative information Direct analysis of drug distribution Investigation of drug penetration	[84,85]
RSOM	1.5–5 mm	Light absorption	Qualitative Non-invasive	Longer scan times	Observation of vascular structures	[86]
3D OrbiSIMS	1–2 µm	Collection of secondary ions	Quantitative High-resolution	Can only analyze standard compounds	Examination of endogenous species	[87]

**Table 4 pharmaceutics-15-01201-t004:** Natural products for hair regrowth.

Herbal Bioactive	Plant Origin	Mechanism of Action	Study Model	Dosage	Conclusions	References
Tectoridin	*Rhizoma belamcandae*	Activating Wnt/β-catenin signaling in human dermal papilla cells	Human follicular DPCs/mouse vibrissae organ	Cells: 3, 10, 20, and 50 µM Mouse vibrissae organ: 50 or 100 µM	Tectoridin promoted hair growth in a dose-dependent manner	[90]
Cepharanthine	*Stephania cephalantha*	Stimulating the growth of human DPCs (hDPCs), significantly increasing the expression of VEGF and the concentration of intracellular Ca^2+^, as well as increasing the expression of HIF-1α and HIF-2α and HIF-responsive genes in hDPCs	hDPCs	0.625/1.25/2.5 μg/mL	Cepharanthine promoted the proliferation of hDPCs and increased the expression of VEGF	[91,92]
Astragaloside IV	*Astragalus membranaceus*	Blocking the Fas/Fas L-mediated apoptotic pathway, blocking procaspase-8, and inhibiting caspase-3 and procaspase-9 activities, thus causing the downregulation of Bax and p53, upregulation of Bcl-2 and Bcl-xL, and inhibition of NF-κB and IkB-α phosphorylation, accompanied by a decrease in the levels of three MAPKs: ERK, SAPK/JNK, and p38	Seven-week-old female C57BL/6 mice (18–20 g)	1 mM or 100 mM	Astragaloside IV inhibited apoptosis-regression catagen in hair follicles via the Fas/Fas L-mediated cell death pathway, the terminal differentiation of hair keratinocytes, and the levels of regeneration factors and cytokines, leading to hair regeneration.	[93]
Acankoreoside J	*Acanthopanax koreanum*	Increasing nuclear β-catenin levels, upregulating cyclin D1, cyclin E, and CDK2, and downregulating p27kip1 in DPCs, thus promoting DPC proliferation	DPCs male Wistar rats (23 days old)	Rat vibrissa dermal papilla cells were treated with acankoreoside J (AK10) at 0.1, 0.2, 1, and 2 µM for 4 days; rat vibrissa follicles were treated with acankoreoside J (AK10) at 0.1, 1, and 10 µM for 21 days	The proliferation of DPCs and the regeneration of hair fibers in rat vibrissa follicles were increased. Hence, acankoreoside J (AK10) might have therapeutic potential for the promotion of hair regeneration	[94]
3-Deoxysappanchalcone	*Caesalpinia sappan* L. (*Leguminosae*)	Stimulating hair regeneration likely by inducing the proliferation of follicular DPCs via the modulation of Wnt/β-catenin and STAT signaling	Human hair follicle DPCs; Seven-week-old female C57BL mice	Two hundred microliters of 3-DSC (3 mM) were applied twice daily for 15 days	The proliferation of HHDPCs and mouse hair regeneration increased in vivo	[95]
Cedrol	*Platycladus orientalis*	\	Six-week-old C57BL/6 mice (body weight 18–20 g)	Different doses (10, 20, 30) mg/mL of cedrol were applied topically to the test area (100 mL) once a day, for 21 days.	Cedrol promoted hair regeneration in a dose-dependent manner	[96,97,98,99]
Baicalin	*Scutellaria baicalensis*	Activating Wnt/β catenin signaling, enhancing ALP expression and activity in human DPCs, and increasing the expression of IGF-1 and VEGF	Eight-week-old female C57BL/6 mice	Different doses (50 and 100 μM) in 50% ethanol 1(50 μL) were applied topically every day for approximately 5 weeks	Baicalin promoted hair regeneration by inducing anagen from telogen in mice	[100,101]
Physcion	*Polygonum multiflorum*	Inhibiting 5α-reductase	Seven-week-old C57BL/6 mice	100 μL of Physcion solution (5 mg/mouse/day or 2 mg/mouse/day) for 28 days	Physcion could shorten the time of dorsal skin darkening and hair regeneration, improve hair follicle morphology, and significantly increase hair follicle count	[102]
Quercetrin	*Hottuynia cordata*	Activating the MAPK/CREB signaling pathway	hDPCs/mouse vibrissae organ	Cells: 10 and 100 nM, 1 μM Mouse vibrissae organ: 5 and 10 µM	Increased the expression of growth factors such as bFGF, KGF, PDGF-AA, and VEGF, promoting cell proliferation, and increased the phosphorylation of Akt, Erk, and CREB in cultured hDPCs, stimulating hair growth	[103]
Quercetin	*Rutin*	Reducing HSP70 levels in skin lesions of mouse models of heat-induced alopecia	Human umbilical vein endothelial cells (HUVECs) and human hair follicle DPCs (HHDPCs) Female C3H/HeJ breeders (approximately 6 months old)	Mice were injected subcutaneously with 100 μL of 10 μM quercetin in 10% dimethyl sulfoxide (DMSO) in phosphate-buffered saline (PBS) once daily for 8 days. Mice were monitored for 6 weeks for hair regeneration.	Quercetin induced hair regeneration in preexisting alopecic lesions in C3H/HeJ mice with spontaneous AA	[104,105,106,107]
EGCG	Green tea	Downregulating the apoptosis of follicular epithelial cells by upregulating phosphorylated Erk and Akt and increasing the Bcl-2/Bax ratio Androgen metabolism: inhibiting 5a-reductase and repressing the transcription of the androgen receptor (AR) gene	B6CBACF1⁄J female mice (7 weeks old)	Topical application of 100 µL 10% EGCG; T (2 mg/day) was injected intradermally, once daily for 5 days a week, for 12 weeks. The experiment lasted 12 days	EGCG stimulated human hair regeneration through dual proliferative and anti-apoptotic effects on DPCs	[108,109]
Forsythiaside-A	*Forsythia suspensa*	Reducing the expression of TGF-β2, caspase-9, and caspase-3	HHDPCs, human keratinocytes (HaCaTs)	Test group: injected with DHT (30 mg/EA) and orally administered 100 μL of forsythiaside-A (1%) once a day for 35 days	Forsythiaside-A prevented apoptosis, which was induced by DHT and delayed the entry of catagen in androgenic alopecia	[110]

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
