# Peer review of "Innovative Strategies for Hair Regrowth and Skin Visualization"

_pharmaceutics, 2023, doi:10.3390/pharmaceutics15041201_

Round 1
Reviewer 1 Report
Review of Manuscript Entitled “Innovative Strategies for Hair Growth and Skin Visualization
This reviewer will begin by noting that I am not an expert in hair biology. This is important as another referee must be included who is more nuanced in the cell biology of hair regrowth to review the data presented in Section 3 and Table 1.
The manuscript entitled “Innovative strategies for hair regrowth and skin visualization” is a very extensive review of the latest technologies that are being examined for hair improvements and regrowth. The paper is quite well written with very few typos (see notes below). In some respects, the paper may be an over ambitious attempt to cover too many topics as the paper tries to cover emerging hair growth technologies, skin visualization technologies and herbal treatments for hair growth. It is this referee’s thought that the section on skin visualization could almost be another paper, perhaps in the same journal. But I understand why the authors are putting it with the paper on hair regrowth.
It may also be helpful if the authors, who appear to be skilled in technologies related to hair regrowth, consider providing a section in the paper addressing the legal aspects of the claims around hair regrowth as it pertains to various parts of the world. For instance, in the US, hair growth or regrowth claims are considered drug claims, while in parts of Korea, for instance, they may be considered cosmetic claims. A short review of global regulatory understanding would be very helpful in this kind of comprehensive review.
The following are some general comments that will help improve the manuscript.
The authors need to expand some discussion on the difference between the human hair follicle and the mouse/rat hair follicle. So much of the work that is being done on hair regrowth is done on either cultured hair follicles or on mice/rat skin. However, as is often the case, the technologies that work in these testing methods fail to influence the human scalp. This may be due to several reasons including the fact that physiological pathways of hair growth in humans are different than mice/rats and more importantly, the ability to get active ingredients to penetrate to the hair bulge where the stems cells are located and to the hair bulb where the dermal papillae are located is difficult. The authors note this with their section on microneedling but more emphasis on these difficulties should be considered. Also, some discussion on eyelash regrowth might be interesting (but not essential) as this is another area where understanding of hair biology has become a target with the recent launch of Latisse.
It is the reviewer’s recommendation that the authors modify Figure 1 to not show so much the structure of the human skin, but instead offer two clear images, one of a human hair follicle and one of a mouse hair follicle and add a written section that looks more deeply at the differences. In this section, the authors can note that while the follicle structures are different, various physiological pathways to hair grow are similar. Another concern is that the images that the authors appear to have gained copyright approval to use are small in many cases making seeing important details difficult. These images should be improved so readers can see them in greater detail.
This reviewer was a little surprised that the authors appeared to have not included discussion on JAK inhibitors that are currently showing very interesting promise for improving hair regrowth in people suffering from alopecia areata. While this is an autoimmune-driven problem, there has been some tremendous strides made to improve hair regrowth in people suffering from this condition and it should be addressed in a comprehensive review such as this paper.
The following are more specific problems that were noted that need to be corrected.
· Page 3, Line 103: “…like bacteria and viruses and toxic substances from [entering] the body.”
· Page 4, Line 114: “…the invaginating part [p] dermal…” remove the p
· Page 4, Figure 1: The figure should be modified as noted above and including direct visualization of such structures as the Huxley layers and Henle’s layers.
· Page 13, Line 384: The sentence on my copy of the paper ends with “…how growth factors affect HFs have n…”. The next sentence on Page 14 does not conclude the sentence on Page 13. Some section of the paper appears to be missing.
· Page 21, Lines 604-605: This sentence which states; “However, there are various differences between mouse and human skin” supports my summary that the authors need to expand this discussion more thoroughly and not just for skin, but more importantly for the differences between mouse and human hair follicles.
· Page 29, Lines 768-769: The sentence; “Traditional Chinese medicine provides a valuable method for treating hair loss.” Is unsubstantiated in this referee’s opinion. There are many Chinese who are bald and treatments that are not clinically tested boarder on pseudoscience and should be very carefully reviewed with this in mind.
· Table 4. Extending onto the discussion above, Table 4 is the author’s summary of herbal supplements that may, or may not, improve hair growth. The data in Table 4 is principally generated from cell cultures. And, while it is nice to present scientific suggestions of biological pathways to hair improvements, i.e., “Mechanisms of Action”, it is another to correlate these to actual clinically tested hair growth. The authors would be better served here in going into the literature and summarizing clinical studies that have examined hair regrowth. For instance, there are some clinical studies out of Japan that demonstrate that the addition of Adenosine to the scalp can improve hair regrowth [see for example: Lisztes E et al – J Invest Dermatol 2019, https://pubmed.ncbi.nlm.nih.gov/31730764/]. Unfortunately, by adding the section on herbal supplements, the author’s have taken what appears to be a very comprehensive, science-based review, and diminished it by promoting pseudoscience. If there are clinical studies that support certain herbal supplements either taken orally or applied topically, this would be much more interesting to the readers than cell culture data that is, for all intents and purposes, meaningless.
Author Response
Many thanks for your very useful comments and suggestions to our manuscript.
- This reviewer will begin by noting that I am not an expert in hair biology. This is important as another referee must be included who is more nuanced in the cell biology of hair regrowth to review the data presented in Section 3 and Table 1.
Response: Thank you for your suggestions.
- The manuscript entitled “Innovative strategies for hair regrowth and skin visualization” is a very extensive review of the latest technologies that are being examined for hair improvements and regrowth. The paper is quite well written with very few typos (see notes below). In some respects, the paper may be an over ambitious attempt to cover too many topics as the paper tries to cover emerging hair growth technologies, skin visualization technologies and herbal treatments for hair growth. It is this referee’s thought that the section on skin visualization could almost be another paper, perhaps in the same journal. But I understand why the authors are putting it with the paper on hair regrowth.
Response: Thank you for your suggestions. To assess drug penetration and retention in specific skin structures, skin visualization techniques are needed. These techniques can provide images of drug accumulation in the skin. Indeed, some skin visualization techniques can also quantify the amount of drug accumulation. These imaging techniques are specific and sensitive and provide qualitative or quantitative information using invasive or non-invasive imaging approaches.
- It may also be helpful if the authors, who appear to be skilled in technologies related
to hair regrowth, consider providing a section in the paper addressing the legal aspects of the claims around hair regrowth as it pertains to various parts of the world. For instance, in the US, hair growth or regrowth claims are considered drug claims, while in parts of Korea, for instance, they may be considered cosmetic claims. A short review of global regulatory understanding would be very helpful in this kind of comprehensive review.
Response: Thank you for your suggestions. We have added related claims in Part 6 “Claims for hair regrowth”according to your comments. Details are as follows:
Page 30-31, Line 803-815, “Nowadays, different countries have different regulations and laws for hair regrowth products. In China, hair regrowth products are classified as cosmetics. These products need to be reported to the State Food and Drug Administration (SFDA), and an approval number needs to be obtained. In Japan, hair regrowth products are classified as cosmetic, quasi-pharmaceutical, and pharmaceutical products according to their composition, claims, and efficacy. However, hair regrowth products are classified as pharmaceutical products in the USA and are subject to FDA approval, which encom-passes the product name, active ingredient, and concentration. For example, rogaine — a star hair product classified as a medicine in the USA — contains MXD as its main active ingredient. In China, MXD is classified as a drug that can bought in pharmacies but cannot be used in cosmetics. To gain FDA approval, hair regrowth products must provide substantial data supporting their efficacy for hair regrowth. When using hair regrowth products, both the instructions for application/use and side effects must be noted carefully.”
- The authors need to expand some discussion on the difference between the human hair follicle and the mouse/rat hair follicle. So much of the work that is being done on hair regrowth is done on either cultured hair follicles or on mice/rat skin. However, as is often the case, the technologies that work in these testing methods fail to influence the human scalp. This may be due to several reasons including the fact that physiological pathways of hair growth in humans are different than mice/rats and more importantly, the ability to get active ingredients to penetrate to the hair bulge where the stems cells are located and to the hair bulb where the dermal papillae are located is difficult. The authors note this with their section on microneedling but more emphasis on these difficulties should be considered.
Response: Thank you for your suggestions. Human and mouse HFs have the same basic structures, but we have discussed differences aspects between human and mouse HFs in Part 2 “The structure of skin and hair follicle”. Details are as follows:
Page 4, Line 135-144, “Human and mouse HFs have the same basic structure, including the bulge, ORS, IRS, isthmus, and infundibulum. They also possess the same principal cell types and undergo hair cycling, anagen phases, catagen phases, and telogen phases. However, some differences exist between human and mouse HFs. First, the anagen phase of human HFs can last for several years, but the anagen phase of mouse skin HFs only lasts for 2–3 weeks. Second, the markers of epithelial HFSCs are different. CD34, a bulge cell marker in the mouse, is not expressed in human bulge cells but is instead expressed in ORS cells in human HFs. Third, the HF cycle in humans is asynchronous, while that in mice is synchronized. Despite these differences, murine models remain important for studying human HF cycling.”
In the “Microneedle” parts, we will add some information to discuss the use of microneedle between human and mouse skin. Details are as follows:
Page 16, Line 485-490, “Further, differences in HF length between mice and humans also need to be considered. Human HFs are significantly larger, reaching lengths of 5 mm, whereas mouse follicles are only 1-mm long. Microneedles can provide effective drug delivery to mouse HFs due to their length. However, microneedles may not deliver drugs to the target after application on human skin because their length may be too short to reach the HF site.”
- Also, some discussion on eyelash regrowth might be interesting (but not essential) as this is another area where understanding of hair biology has become a target with the recent launch of Latisse.
Response: Thank you for your suggestions. It seems that discussion on eyelash regrowth might be interesting, but the examples we had listed in this review were emphasized on the innovative strategies for hair regrowth on the back skin of mice. If we discuss on eyelash regrowth, it might involve different innovative strategies and different mechanisms of hair regrowth, but we will continue to focus the related references of eyelash regrowth.
- It is the reviewer’s recommendation that the authors modify Figure 1 to not show so much the structure of the human skin, but instead offer two clear images, one of a human hair follicle and one of a mouse hair follicle and add a written section that looks more deeply at the differences. In this section, the authors can note that while the follicle structures are different, various physiological pathways to hair grow are similar.
Response: Thank you for your suggestions. We have replaced the Figure 1 according to your suggestions. We have discussed the differences in Part 2 “The structure of skin and hair follicle”. Details are as follows:
Page 4, Line 135-144, “Human and mouse HFs have the same basic structure, including the bulge, ORS, IRS, isthmus, and infundibulum. They also possess the same principal cell types and undergo hair cycling, anagen phases, catagen phases, and telogen phases. However, some differences exist between human and mouse HFs. First, the anagen phase of human HFs can last for several years, but the anagen phase of mouse skin HFs only lasts for 2–3 weeks. Second, the markers of epithelial HFSCs are different. CD34, a bulge cell marker in the mouse, is not expressed in human bulge cells but is instead expressed in ORS cells in human HFs. Third, the HF cycle in humans is asynchronous, while that in mice is synchronized. Despite these differences, murine models remain important for studying human HF cycling.”
- Another concern is that the images that the authors appear to have gained copyright approval to use are small in many cases making seeing important details difficult. These images should be improved so readers can see them in greater detail.
Response: Thank you for your suggestions. We have improved the resolution of all the Figures.
- This reviewer was a little surprised that the authors appeared to have not included discussion on JAK inhibitors that are currently showing very interesting promise for improving hair regrowth in people suffering from alopecia areata. While this is an autoimmune-driven problem, there has been some tremendous strides made to improve hair regrowth in people suffering from this condition and it should be addressed in a comprehensive review such as this paper.
Response: Thank you for your suggestions. We have discussed the JAK inhibitor on the AA in the “Introduction” part. Details are as follows:
Page 2, Line 46-49, “AA is an autoimmune disease clinically characterized by small, bald patches on the head. Several clinical trials have reported the use of Janus kinase (JAK) inhibitors, including ruxolitinib, tofacitinib, and baricitinib, for AA treatment.”
9.The following are more specific problems that were noted that need to be corrected.
Page 3, Line 103: “…like bacteria and viruses and toxic substances from [entering] the body.”
Page 4, Line 114: “…the invaginating part [p] dermal…” remove the p
Page 4, Figure 1: The figure should be modified as noted above and including direct visualization of such structures as the Huxley layers and Henle’s layers.
Page 13, Line 384: The sentence on my copy of the paper ends with “…how growth factors affect HFs have n…”. The next sentence on Page 14 does not conclude the sentence on Page 13. Some section of the paper appears to be missing.
Page 21, Lines 604-605: This sentence which states; “However, there are various differences between mouse and human skin” supports my summary that the authors need to expand this discussion more thoroughly and not just for skin, but more importantly for the differences between mouse and human hair follicles.
Page 29, Lines 768-769: The sentence; “Traditional Chinese medicine provides a valuable method for treating hair loss.” Is unsubstantiated in this referee’s opinion. There are many Chinese who are bald and treatments that are not clinically tested boarder on pseudoscience and should be very carefully reviewed with this in mind.
Table 4. Extending onto the discussion above, Table 4 is the author’s summary of herbal supplements that may, or may not, improve hair growth. The data in Table 4 is principally generated from cell cultures. And, while it is nice to present scientific suggestions of biological pathways to hair improvements, i.e., “Mechanisms of Action”, it is another to correlate these to actual clinically tested hair growth. The authors would be better served here in going into the literature and summarizing clinical studies that have examined hair regrowth. For instance, there are some clinical studies out of Japan that demonstrate that the addition of Adenosine to the scalp can improve hair regrowth [see for example: Lisztes E et al – J Invest Dermatol 2019, https://pubmed.ncbi.nlm.nih.gov/31730764/]. Unfortunately, by adding the section on herbal supplements, the author’s have taken what appears to be a very comprehensive, science-based review, and diminished it by promoting pseudoscience. If there are clinical studies that support certain herbal supplements either taken orally or applied topically, this would be much more interesting to the readers than cell culture data that is, for all intents and purposes, meaningless.
Response: Thank you for your suggestions. We have revised according to your comments. Details are as follows:
We have added the word “entering” on Page 4, Line 119 “…like bacteria and viruses and toxic substances from [entering] the body.”
We have deleted ‘p’ on Page 4, Line 120: “…invaginating dermal regions…”
Page 4, Figure 1: We have divided into two parts of human and mouse hair follicle.
We have added the missing information on Page 15, Line 417: “…how growth factors affect HFs have not been clarified”.
Page 22, Lines 643-645: This sentence which states “However, there are various differences between mouse and human skin” have been replaced by “However, there are various differences between mouse and human HFs. For examples, the anagen phase is only 2–3 weeks in mice but can last for several decades in humans. Further, markers of epithelial HFSCs are also different.”
Page 32, Line 825-826, “Traditional Chinese medicine provides a valuable method for treating hair loss.” have changed into “Traditional Chinese medicine has broadened the ideas for treating hair loss.”
Table 4 has been supported by the addition of literatures on the basis of the original, and the added literatures are mainly about the application of the natural products in preclinical studies and reported clinical studies in hair loss. Unfortunately, the clinical efficacy of natural products applied to humans has few report, most of the studies focus on the preclinical studies, mainly to explore whether natural products have hair loss efficacy and mechanism research, on the basis of known exact efficacy, further by preparing liposomes or hydrogels or combined with adjuvant therapy to enhance skin permeability. Hence, we have added related clinical trials on Part 7 “Natural products for hair regrowth”, Details are as follows:
Page 36, Line 876-889, “Preclinical studies are a key component of research on the effect of herbs and their active constituents on hair growth. In comparison to the high number of in vitro and in vivo preclinical studies, there are few clinical trials describing the influence of herbs and their active constituents as well as that of polyherbal formulations on hair growth. A small clinical study conducted with 3 volunteers demonstrated that 10% EGCG can promote hair growth by prolonging the anagen phase. In one case series of 14 patients with AA, Morita et al. reported that the topical administration of the immunotherapy agent squaric acid dibutylester (SADBE) for a mean of 6.9 months led to no or poor hair regrowth. However, the subsequent administration of a combination of topical SADBE treatment plus oral CEP for a mean duration of 7.6 months resulted in satisfactory hair regrowth in six of the 14 cases. Meanwhile, attempts have been made to develop new pharmaceutical dosage forms. Given that cedarferol is known to treat hair loss, a cedarferol nanoemulsion and cedarferol cream were applied to the skin in an animal model and were found to increase drug permeability and significantly promote hair regrowth.”

Reviewer 2 Report
I agree "hair loss is worth studying. If we take additional time to learn from HFs and understanding the processes of repair and regeneration, more and more potential strategies for hair regrowth will be discovered. Fresh and exciting strategies for hair loss management will be imminent in the future."
Author Response
Many thanks for your very useful comments and suggestions to our manuscript. We have modified the manuscript in blue text according to your comments, and the detailed point-to-point responses are listed below:
I agree "hair loss is worth studying. If we take additional time to learn from HFs
and understanding the processes of repair and regeneration, more and more potential strategies for hair regrowth will be discovered. Fresh and exciting strategies for hair loss management will be imminent in the future."
Response: Thank you for your suggestions. Related information has been added in part “Structure of the skin and hair follicles.” Details are as follows:
Page 4, Line 129-135, “Hair bulbs are surrounded by dermal fibroblasts, and fibroblasts constitute dermal papillae (DPs), which possess the ability to induce HF renewal. However, how growth factors affect HFs has not been clarified. The IRS is made up of three layers, including the root sheath, Huxley’s layer, and Henle’s layer, and the three-layered structure of the IRS decides the shape of the hair shaft. Outside the IRS lies the ORS, which encompasses the whole IRS and hair shaft. The outermost layer of a HF is the CTS, which consists of several layers of fibroblast cells.”

Reviewer 3 Report
The authors did a nice job reviewing the various known modalities for stimulating hair growth. The visuals presented were appropriate and helpful to me as I reviewed each category. The biochemistry and physiology for hair growth is nicely reviewed.
Three areas that were not discussed that should be included in this review paper are:
1- Information discussing the benefits of wounding with microneedling and how injuries to the stem cells in the upper hair follicle brings on hair growth. The authors limited the benefits of microneedling to the ability to enhance drug delivery while today, the use as an injury tool is more prevalent and fairly effective.
2- The authors also failed to mention Sulfotransferases, the enzyme that enhances the conversion of Minoxidil to Minoxidil Sulfate, the active form of minoxidil for hair growth. Sulfotransferases is found in both the liver and and in the 'outer root hair sheath' of the hair follicle and when present in the hair follicle, it means that the person will be a responder to minoxidil. Only about 40% of people are reported to be responders to topical minoxidil but the response goes much higher when minoxidil is taken orally because Sulfotransferase works in the liver producing Minoxidil Sulfate systemically and therefore making it more available to the hair follicle.
3- I also noted the omission in the use of liposomal chemistries used with finasteride and minoxidil. These chemistries are presently used to counter the side effects of these drug. There is a considerable amount of published data in the literature for the use of liposomes in delivering the drug to the scalp, thereby limiting their systemic effects.
Author Response
Many thanks for your very useful comments and suggestions to our manuscript. We have modified the manuscript in blue text according to your comments, and the detailed point-to-point responses are listed below:
1.Information discussing the benefits of wounding with microneedling and how injuries to the stem cells in the upper hair follicle brings on hair growth. The authors limited the benefits of microneedling to the ability to enhance drug delivery while today, the use as an injury tool is more prevalent and fairly effective.
Response: Thank you for your suggestions. We have added related information in “Microneedles” part. Details are as follows:
Page 16, Line 481-485, “Accordingly, MNs can not only improve the efficiency, accuracy, and effectiveness of drug delivery but also stimulate re-epithelialization to promote hair regrowth. However, the injury caused by microneedles cannot be ignored. Transient pain and mild erythema are commonly reported as adverse effects of microneedles.”
2.The authors also failed to mention Sulfotransferases, the enzyme that enhances the conversion of Minoxidil to Minoxidil Sulfate, the active form of minoxidil for hair growth. Sulfotransferases is found in both the liver and and in the 'outer root hair sheath' of the hair follicle and when present in the hair follicle, it means that the person will be a responder to minoxidil. Only about 40% of people are reported to be responders to topical minoxidil but the response goes much higher when minoxidil is taken orally because Sulfotransferase works in the liver producing Minoxidil Sulfate systemically and therefore making it more available to the hair follicle.
Response: Thank you for your suggestions. We have added related information in “Introduction” part. Details are as follows:
Page 3, Line 62-68, “MXD needs to be converted into its active derivative, minoxidil sulphate, and this re-action is catalyzed by sulfotransferases. The sulfotransferase 1 (SULT1) family of enzymes is expressed in the lower sheath of the HF and the liver. It converts the pro-drug MXD to its active form, minoxidil sulfate, in the outer root sheath (ORS) of HFs. SULT1A1 is the predominant isoenzyme responsible for the sulfonation of MXD in HFs. The expression of sulfotransferase in the scalp is different in different individuals, ex-plaining the heterogeneity in clinical responses to MXD therapy.”
3.I also noted the omission in the use of liposomal chemistries used with finasteride and minoxidil. These chemistries are presently used to counter the side effects of these drug. There is a considerable amount of published data in the literature for the use of liposomes in delivering the drug to the scalp, thereby limiting their systemic effects.
Response: Thank you for your suggestions. We have added related information in “Introduction” part. Details are as following:
Page 3, Line 66-73, “The expression of sulfotransferase in the scalp is different in different individuals, ex-plaining the heterogeneity in clinical responses to MXD therapy. In some studies, the skin permeability and retention of FIN and MXD were enhanced using lipo-some-based delivery systems. For example, the use of FIN has been limited because its systemic administration can cause sexual dysfunction. To solve this problem, DMSO-modified liposomes were prepared for the topical delivery of FIN. The perme-ation capacity of DMSO could promote FIN delivery to HFs, mitigating the adverse effects of systemic administration.”

Reviewer 4 Report
This is a well-written comprehensive paper on treatments for hair-loss.
I have a major comment:
It is not very clear if all the described mechanisms and treatments apply to all types of hair loss or only AGA. This should be clear and the title should reflect the correct type of alopecia. AGA is not actually a hair loss but hair-miniaturisation. This means that histological findings and even clinical findings will be different between AA, AGA and cicatricial alopecias, not to mention on cell- and cytokine levels. The way the paper is presented it may seem that all forms of alopecia have the same ethiopathologies, diagnostics and treatments. Please use correct wording to avoid misunderstanding. Testosterone will not have the same effect on AA as in AGA. This is not at all explained.
Minor:
- Images of higher resolution are recommended
- All abbreviations should be given in full first time mentioned even if obvious. They should also be explained in legends/footnotes in tables even if described in the text.
Author Response
Many thanks for your very useful comments and suggestions to our manuscript. We have modified the manuscript in blue text according to your comments, and the detailed point-to-point responses are listed below:
1.It is not very clear if all the described mechanisms and treatments apply to all types of hair loss or only AGA. This should be clear and the title should reflect the correct type of alopecia. AGA is not actually a hair loss but hair-miniaturisation. This means that histological findings and even clinical findings will be different between AA, AGA and cicatricial alopecias, not to mention on cell- and cytokine levels. The way the paper is presented it may seem that all forms of alopecia have the same ethiopathologies, diagnostics and treatments. Please use correct wording to avoid misunderstanding. Testosterone will not have the same effect on AA as in AGA. This is not at all explained.
Response: Thank you for your suggestions. Details are as follows:
Page 2, Line 41-46, “the most common is AGA, a chronic and progressive disease that is also called male pattern baldness. In AGA patients, dermal papilla cells (DPCs) express high levels of androgen receptors (ARs), which increases their sensitivity to androgens. When the androgen testosterone binds to an AR, it is converted into dihydrotestosterone (DHT) in the cytoplasm of DPCs. This reaction is catalyzed by the enzyme type II 5α-reductase (SRD5A2).”
Page 2, Line 46-49, “AA is an autoimmune disease clinically characterized by small, bald patches on the head. Several clinical trials have reported the use of Janus kinase (JAK) inhibitors, including ruxolitinib, tofacitinib, and baricitinib, for AA treatment.”
2.Images of higher resolution are recommended.
Response: Thank you for your suggestions. We have improved the resolution of all the figures.
3.All abbreviations should be given in full first time mentioned even if obvious. They should also be explained in legends/footnotes in tables even if described in the text. Response: Thank you for your suggestions. We have checked our abbreviations carefully according to your suggestions.

Reviewer 5 Report
2023-03-17
Review of pharmaceutics-2241070
Dear Authors let me state that your review covers very important subject. But there are a lot of minor points that need to be addressed before your submission reaches the readers:
1. Figure 2. (a) – on subpanel iii) designations are invisible size wise; (b) – text and scale bars on subpanels is invisible both colorwise and sizewise; (c) the on-hand writing isn’t visible
2. Figure 3 – Low visibility both sizewise and colorwise. Remove black background from dark blue images. Do not put panels in the row, but beneath each other
3. Figure 4. Increase size of panels c, d, and f. Just simply increase the size
4. Figure 5. Panel d - Remove black background (from dark blue images especially).
5. Figure 6. Panel e - Remove black background (from dark blue images especially).
6. Figure 7. Panels b-d. Clarify scale bars size since it is invisible
7. Figure 8. Panels a-d. Clarify scale bars size since it is invisible. Panel h – formula is invisible
8. Figure 9. Panels c,e,d,f,i-l. Clarify scale bars size since it is invisible. Text on all panels is too small and blurry.
9. Figure 10 requires scale bar(s)
Conclusion: accept after Figures correction
Author Response
Many thanks for your very useful comments and suggestions to our manuscript. We have modified the manuscript in blue text according to your comments, and the detailed point-to-point responses are listed below:
- Figure 2. (a) – on subpanel iii) designations are invisible size wise; (b) – text and scale bars on subpanels is invisible both colorwise and sizewise; (c) the on-hand writing isn’t visible
- Figure 3 – Low visibility both sizewise and colorwise. Remove black background from dark blue images. Do not put panels in the row, but beneath each other
- Figure 4. Increase size of panels c, d, and f. Just simply increase the size
- Figure 5. Panel d - Remove black background (from dark blue images especially).
- Figure 6. Panel e - Remove black background (from dark blue images especially).
- Figure 7. Panels b-d. Clarify scale bars size since it is invisible
- Figure 8. Panels a-d. Clarify scale bars size since it is invisible. Panel h – formula is invisible
- Figure 9. Panels c,e,d,f,i-l. Clarify scale bars size since it is invisible. Text on all panels is too small and blurry.
- Figure 10 requires scale bar(s)
Response: Thank you for your suggestions. We have obtained the figure from the press, the background of fluorescence images could not modify, but we have improved the resolution of each figure and added the scale bars. Figure 10 is to visualize the vasculature of the subcutaneous, but the reference do not provide the scale bars.

Round 2
Reviewer 5 Report
Dear authors, Let me cjmment your general response:
"Thank you for your suggestions. We have obtained the figure from the press, the background of fluorescence images could not modify, but we have improved the resolution of each figure and added the scale bars."
As for me, I do not consider your response as solid for two resons:
1. I should think about convenience to the readers
2. I discuss namely your review, not somebody's else
As for me, I appreciate modifications that you did so far and exclude the acceptable parts from further consideration. Let me now list the spots, where letters SHOULD be enlarged. You can place enlarged text in the boxes nearby. Also , you could proportionally expand the Figure 30% in length; and you can place panels and subpanels in a column instead of row. So, let's comment the Figures again:
- Figure 2. (a) – on subpanel i) small text in unreadable; on subpanel ii) white text is unreadable; on subpanel iii) inside the cirle designations text is invisible size; (b) – text and scale bars on subpanels is invisible both colorwise and sizewise.
- Figure 3 – Low visibility both sizewise and colorwise. Remove black background from dark blue images. Do not put panels in the row, but beneath each other. SCALEBARS!!!!!
- Figure 4. Increase size of text on subpanel f. Just simply increase the size
- Figure 5. Panel d - SCALEBARS!!!.
- Figure 6. Just simply increase the size. Panels c,eg - What is the size of scalebars?
- Figure 7d. Clarify scale bars size since it is unreadable
- Figure 8. Panels a-d. Clarify scale bars size since it is invisible. Panel h – formula is invisible. The description of vertical axel on panel h is blurry and invisible.
- Figure 9. Panel d - the descriptions of skin layers are blurry and invisible.
Please fix it for the sake of readers' convenience
Author Response
Many thanks for your very useful comments and suggestions to our manuscript. We have modified the manuscript in blue text according to your comments, and the detailed point-to-point responses are listed below:
Reviewers' comments:
As for me, I do not consider your response as solid for two reasons:
- I should think about convenience to the readers
- I discuss namely your review, not somebody's else
As for me, I appreciate modifications that you did so far and exclude the acceptable parts from further consideration. Let me now list the spots, where letters SHOULD be enlarged. You can place enlarged text in the boxes nearby. Also, you could proportionally expand the Figure 30% in length; and you can place panels and subpanels in a column instead of row. So, let's comment the Figures again:
- Figure 2. (a) – on subpanel i) small text in unreadable; on subpanel ii) white text is unreadable; on subpanel iii) inside the cirle designations text is invisible size; (b) – text and scale bars on subpanels is invisible both colorwise and sizewise.
- Figure 3 – Low visibility both sizewise and colorwise. Remove black background from dark blue images. Do not put panels in the row, but beneath each other. SCALEBARS!!!!!
- Figure 4. Increase size of text on subpanel f. Just simply increase the size
- Figure 5. Panel d - SCALEBARS!!!.
- Figure 6. Just simply increase the size. Panels c,eg - What is the size of scalebars?
- Figure 7d. Clarify scale bars size since it is unreadable
- Figure 8. Panels a-d. Clarify scale bars size since it is invisible. Panel h – formula is invisible. The description of vertical axel on panel h is blurry and invisible.
- Figure 9. Panel d - the descriptions of skin layers are blurry and invisible.
Please fix it for the sake of readers' convenience
Response: Thank you for your suggestions. We have revised them according to your comments. Details are as follows:
- Fig 2, we had increased the size of text on subpanel.
- Fig 3, we had increased the size of scalebars, but we could not remove the black background from dark blue images, because it is the dark background of fluorescence image represent that there are no expressions of makers.
- Fig 4, we had increased the size of f.
- Fig 5, we had increased the scalebars.
- Fig 6, we had increased the scalebars.
- Fig 7, we had increased the size of scalebars.
- Fig 8, we had increased the size of scalebars and clarified the formula.
- Fig 9, we had clarified the text of skin layers.
